# Pallidal neuromodulation of the explore/exploit trade-off in decision-making

Ana Luisa de A Marcelino[1,2], Owen Gray[3], Bassam Al-Fatly[1], William Gilmour[3], J Douglas Steele[3], Andrea A Kühn[1,2,4,5,6], Tom Gilbertson[3,7]*

[1]Charité – Universitätsmedizin Berlin, corporate member of Freie Universität Berlin and Humboldt-Universität zu Berlin, Movement Disorder and Neuromodulation Unit, Department of Neurology, Charité Campus Mitte, Berlin, Germany; [2]Berlin Institute of Health at Charité – Universitätsmedizin Berlin, Core Facility Genomics, Berlin, Germany; [3]Division of Imaging Science and Technology, Medical School, University of Dundee, Dundee, United Kingdom; [4]Berlin School of Mind and Brain, Charité - University Medicine Berlin, Berlin, Germany; [5]NeuroCure, Charité - University Medicine Berlin, Berlin, Germany; [6]DZNE, German Centre for Degenerative Diseases, Berlin, Germany; [7]Department of Neurology, Ninewells Hospital & Medical School, Dundee, United Kingdom

*For correspondence:
tgilbertson@dundee.ac.uk

**Abstract** Every decision that we make involves a conflict between exploiting our current knowledge of an action's value or exploring alternative courses of action that might lead to a better, or worse outcome. The sub-cortical nuclei that make up the basal ganglia have been proposed as a neural circuit that may contribute to resolving this explore-exploit 'dilemma'. To test this hypothesis, we examined the effects of neuromodulating the basal ganglia's output nucleus, the globus pallidus interna, in patients who had undergone deep brain stimulation (DBS) for isolated dystonia. Neuromodulation enhanced the number of exploratory choices to the lower value option in a two-armed bandit probabilistic reversal-learning task. Enhanced exploration was explained by a reduction in the rate of evidence accumulation (drift rate) in a reinforcement learning drift diffusion model. We estimated the functional connectivity profile between the stimulating DBS electrode and the rest of the brain using a normative functional connectome derived from heathy controls. Variation in the extent of neuromodulation induced exploration between patients was associated with functional connectivity from the stimulation electrode site to a distributed brain functional network. We conclude that the basal ganglia's output nucleus, the globus pallidus interna, can adaptively modify decision choice when faced with the dilemma to explore or exploit.

## Editor's evaluation

This paper presents valuable data from 18 patients treated with GPi DBS for dystonia using a standard RL task. Their compelling main observation is that DBS reduced the impact of value on evidence accumulation leading to more exploratory choices which was supported by fitting a dynamic decision model to the data. This work will be interesting for scientists working in fundamental and clinical neurosciences.

## Introduction

Flexible adaption of choice behaviour is crucial for decision-making when facing uncertainty. The balance between exploiting familiar options and exploring alternatives with potentially less favorable outcomes, represents a fundamental challenge of adaptive control (*Mehlhorn et al., 2015*). The

neural basis of how humans resolve this dilemma has been advanced significantly over the last decade (*Chakroun et al., 2020*; *Wilson et al., 2014*), but the specific contribution from sub-cortical nuclei that make up the basal ganglia (BG) remains poorly defined (*Cohen et al., 2007*; *Sheth et al., 2011*).

To investigate how humans solve this conflict in reinforcement learning, so-called 'bandit tasks' have been used, where subjects choose between several options associated with different reward probabilities to maximise reward payoff. In a 'reversal-learning' paradigm, the reward probabilities change (i.e. reverse), thus subjects need to constantly evaluate whether choice feedback represents a reversal false alarm (i.e. a lower than expected outcome from the option with the highest reward probability) or a true reversal in the reward probabilities. Even in reversal-learning tasks with low levels of payout uncertainty, human participants make a significant number of choices to the lower value option, rather than the choice with the higher expected value. This leads to the observation of probability matching (*Herrnstein, 1970*), where humans fail to maximise their payoff and adopt a sub-optimal policy which matches their response probabilities to that of the payoff probability. 'Non-greedy' choices, are by definition exploratory, as they are made to the option with the lower expected value (*Daw et al., 2006*). Conversely, 'greedy' choices are made to the option with the highest value (*Findling and Wyart, 2021*; *Costa et al., 2015*; *Izquierdo et al., 2017*) and can be considered to exploit current knowledge of the values of the actions (*Sutton and Barto, 2018*). One possibility is that exploration represents a cognitive strategy for active information seeking, to reaffirm existing beliefs in the relative value of the choices (*Wilson et al., 2014*; *Izquierdo et al., 2017*; *Sutton and Barto, 2018*). Alternatively, it may represent random choice perturbations caused by limitations in the brain's implementation of the decision process leading to decision noise (*Findling et al., 2019*). Either way, the strategies adopted by human participants performing bandit tasks, including reversal-learning (*Costa et al., 2015*; *Costa et al., 2019*; *Costa et al., 2014*), rely on exploiting stability whilst at the same time adapting to volatility by exploring alternative courses of action.

Several observations support a role of the BG circuit in explore-exploit decisions. Computational circuit models of the BG predict that its principal function is action selection – the process where by one action (or decision) is selected over other competing alternatives (*Gurney et al., 2001*; *Bogacz and Gurney, 2007*). An extension of this action selection function is the proposal that it's precision (i.e. how selective the circuit is) can be influenced by the overall excitability of the BG circuit (*Suryanarayana et al., 2019*). In the context of the explore-exploit dilemma, this means the BG may implement a neural 'decision filter', with the bandwidth of this filter being adjusted by neuromodulators, such as dopamine, which influence BG excitability (*Gilbertson and Steele, 2021*). The same models also predict that the excitability of the basal ganglia's principle output nucleus, the globus pallidus interna (GPI) should be a read-out of whether the BG are supporting an exploratory or exploitative decision strategy (*Humphries et al., 2012*; *Chakravarthy et al., 2010*). Consistent with this prediction, the firing rate of GPI neurons in non-human primates encodes the transition from exploratory to exploitative decision making (*Sheth et al., 2011*). Transient decreases in GPI firing observed during learning have been proposed as a potential source of decision variability and could therefore represent a neurophysiological correlate of exploratory (non-greedy) choices observed in bandit tasks.

At a cortical level, functional imaging (fMRI) supports a role for distinct pre-frontal cortical regions in reversal-learning including the orbito-frontal cortex (*Hampshire et al., 2012*; *Remijnse et al., 2005*; *Cools et al., 2002*; *Ghahremani et al., 2010*), ventromedial prefrontal (vmPFC), dorsolateral prefrontal (DLPFC) and anterior cingulate (ACC) cortices. The same pre-frontal regions are also implicated in explore-exploit choices using (so-called 'restless') bandit tasks with high between-trial outcome volatility and associated choice uncertainty (*Chakroun et al., 2020*; *Daw et al., 2006*). Analysis of brain activation patterns in these tasks support distinct cortical regions are involved in the different types of decisions. Consistent with their role within a valuation network, ventromedial prefrontal cortex (vmPFC) and orbitofrontal cortex (OFC) (*Chakroun et al., 2020*; *Daw et al., 2006*; *Findling et al., 2019*; *Badre et al., 2012*; *Tomov et al., 2020*) are activated during exploitative choices. In contrast, exploratory choices activate frontopolar cortex (FPC), dorsolateral prefrontal cortex (DLPFC), anterior insula and anterior cingulate cortex, commensurate with their proposed functions in the encoding of uncertainty, behavioural switching and cognitive control (*Rushworth and Behrens, 2008*; *Boorman et al., 2009*; *Bartolo and Averbeck, 2020*; *Hayden et al., 2011*; *Shenhav et al., 2016*). The striatum, as the BG input nucleus, receives afferents from these same pre-frontal cortical regions forming segregated cortico-striatal-thalamo-cortical circuits which respect the functional anatomy of distinct cortical

circuits for explore-exploit decision making, thus serving as a point of convergence (*Draganski et al., 2008*).

To date, no study has directly addressed the hypothesised role of the GPI in the human approaches to resolving explore-exploit decisions during reversal-learning. Deep Brain Stimulation (DBS) of the GPI is routinely performed as a treatment for isolated generalised dystonia and also in patients with focal and segmental dystonia if botulinum toxin treatment fails (*Volkmann et al., 2014*). Dystonia is a movement disorder characterised by sustained or intermittent muscle contractions causing abnormal postures. In focal dystonia, it is restricted toone region such as the head or neck (*Bhatia et al., 2018*). Patients with dystonia have normal levels of striatal dopamine, intact reward prediction error (RPE) signalling (*Gilbertson et al., 2019*), and exhibit subtle cognitive biases (*Romano et al., 2014*). These include abnormalities of reinforcement learning (RL) such as increased risk taking and delayed flexibility to changes in reward contingency reversal (*Gilbertson et al., 2019*; *Arkadir et al., 2016*). With the caveat of these disease specific RL abnormalities in mind, dystonia patients with chronically implanted DBS-GPI electrodes represent a unique opportunity to test the influence of the GPI and, in turn, as its principle output nucleus, the basal ganglia's role in explore-exploit decision making in humans.

Our working hypothesis was that neuromodulation of the excitability of GPI neurons would modify explore-exploit decisions in a reversal-learning task. As DBS suppresses firing rates in human GPI neurons (*Cleary et al., 2013*; *LafreniereRoula et al., 2010*), we hypothesised that DBS-mediated reduced GPI excitability should drive greater proportion of exploratory choices consistent with previous findings by *Sheth et al., 2011*. To test this hypothesis, we examined choice behaviour in patients with chronically implanted DBS electrodes in both 'ON' and 'OFF' stimulation whilst they executed a two-armed bandit reversal-learning task. Given the more recognised role of the BG in habitual learning of stimulus-response associations (*Piron et al., 2016*; *Redgrave et al., 2010*), we purposely chose a low volatility probabilistic reversal-learning task to engage cortico-sub-cortical circuits in 'model-free' learning (*Daw et al., 2011*). Our choice of this task was also aimed at isolating the specific contribution of the GPI to random exploration. This form of exploration is more closely aligned with the sub-optimal, non-greedy choices identified during pauses in GPI firing in primates (*Sheth et al., 2011*). Exploration can also be 'directed' to the option which strategically aims to minimise uncertainty about a choice that a decision maker has least familiarity with (*Chakroun et al., 2020*). Accordingly, directed exploration relies upon brain regions that encode working memory (*Boorman et al., 2009*) and circuits including the external segment of the globus pallidus (GPe) which contribute to active information seeking to resolve uncertainty (*White et al., 2019*).

By fitting a model of the decision-making process (Reinforcement Learning Drift Diffusion Model: RLDDM), we aimed to test further mechanistic hypotheses regarding latent cognitive effects of DBS neuromodulation on GPI choice arbitration (*Pedersen et al., 2017*). The RLDDM incorporates both decision choices and decision time (DT) information, thus affording a richer interpretation of the mechanisms underlying choice compared to more traditional reinforcement learning models.

We further hypothesised that the influence of GPI-DBS neuromodulation in decision-making should be dependent on the connectivity of the stimulated site with cortical areas implicated in explore–exploit choice behaviour.

## Results

Eighteen patients with isolated dystonia (for clinical details see *Supplementary file 1*) who had chronically implanted DBS electrodes (*Figure 1A*) targeted at the globus pallidus interna (GPI) performed a reversal-learning task (*Figure 1B&C*). The patients were instructed to try and win as many 'vouchers' as possible throughout the duration of the task by choosing one of two options presented to them on a computer screen. Patients were tested with their DBS device turned 'ON' or 'OFF' in a randomised order between the two task blocks. Each block consisted of three, 40-trial sessions where the probability associated with winning a 'voucher' switched from 80%:20% to 20%:80% midway through the second session (*Figure 1B*). Participants performed two versions of this task separated by an interval of 20 min after the initially randomised DBS condition (ON or OFF) had been inverted. The patient's performance was compared with a group of 18 age- and sex-matched healthy controls (HC) performing task block 1. The two blocks differed only by the fractals presented (*Figure 1B*). Participants were not informed about the contingency reversal in the task.

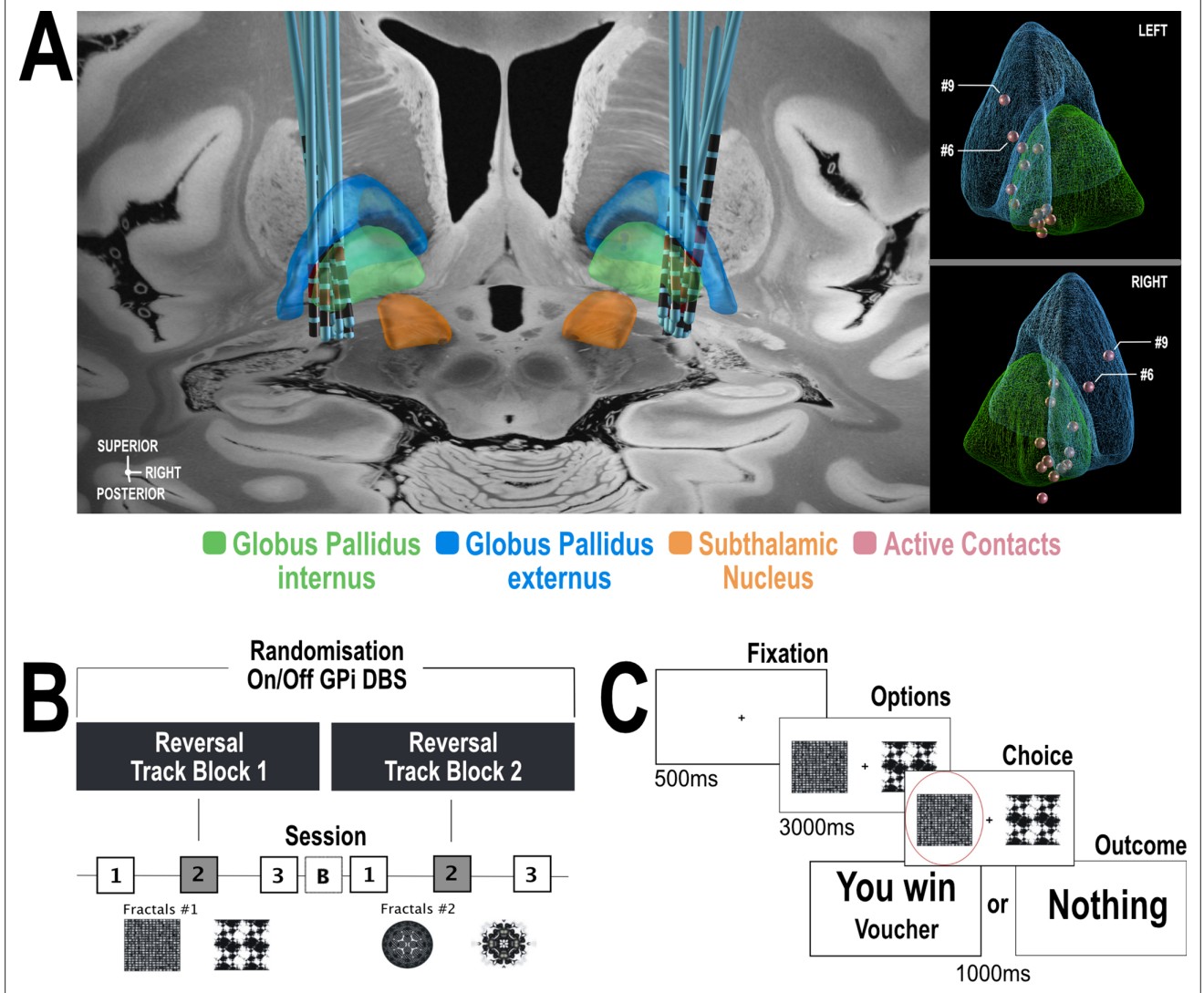

**Figure 1.** General overview of the study design and task. (**A**) Localisation of DBS electrodes was done with Lead-DBS (https://www.lead-dbs.org/) as previously described (; *Horn et al., 2019*). The left panel depicts electrode localisation within the GPI projected to a 7T brain backdrop (*Edlow et al., 2019*). On the right panel, active contacts location is shown in red – only two patients (# 6 and #9) show active contacts within the GPe (blue). (**B**) Each of the two reversal learning task blocks. The blocks differed only in the pair of fractals used as stimulus options. Each reversal learning task block consisted of three 40 trial sessions. Reversal of the probability of receiving a reward occurred half-way through session two. The task was performed once in the OFF- DBS state and once in the ON-DBS state in a counterbalanced manner with a 20 minute break ('B') in between blocks 1 and 2. During this, DBS stimulator was either switched ON or OFF. (**C**) Example of a single trial in the reversal learning task. On each trial, subjects chose either left or right fractal options, which were also counterbalanced, using their left or right hand to press the corresponding keyboard button. The selected cue was then shown surrounded by a red circle (in this example Task Block 1 the left-hand cue is chosen). Subjects were then presented with the outcome of their choice on the next screen, which could be either a reward ('You Win') or zero ('Nothing'). Outcome probabilities of receiving a reward on choosing either fractal were 80%:20%.

The online version of this article includes the following figure supplement(s) for figure 1:

**Figure supplement 1.** Spatial distribution of stimulation volume on a group level.

## GPI DBS enhances exploratory choices without affecting task performance

A fixed effects analysis of performance (number of vouchers won) was conducted, using a two-way repeated measures ANOVA with stimulation state (ON-DBS versus OFF-DBS) and task session as fixed factors. Consistent with the detrimental effect of contingency reversal on performance in session 2, there was a main effect of session on performance ($F_{(2,34)}$ = 6.26, p=0.002, Average rewards:

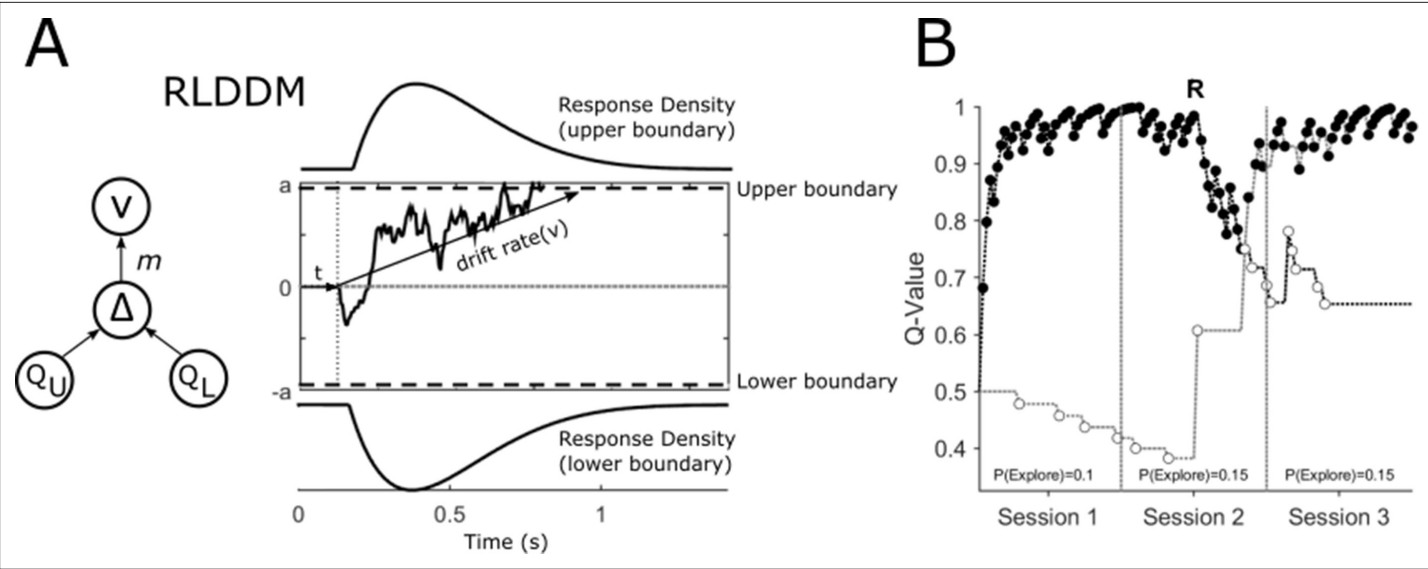

**Figure 2.** Reinforcement Learning Drift Diffusion Model (RLDDM). (**A**) The accumulation of evidence begins at a starting point at 0. The non-decision time is represented by t. Evidence accumulation is represented by a sample path with added Gaussian noise and is gathered until a decision boundary (*a* and -*a*) is reached and a response is initiated. The drift rate *v* determines the rate at which this evidence is accumulated. The extent to which the difference in the expected value of the options of the upper ($Q_U$) and lower response boundaries ($Q_L$) modifies the drift rate is determined by the drift rate scaling parameter, *m*. (**B**) An example of an individual patient's choices (OFF-DBS, Task Block 1) across the three sessions of the task with the expected value of choices represented by the upper $Q_U$ and lower $Q_L$ decision boundaries in black and grey respectively. Closed circles represent 'exploitative' choices where the choice with the highest expected value was chosen. Open circles are 'exploratory', representing the choice of the lower value of the two options. The change in value of the two choices halfway through session 2 of the task reflects the reversal, 'R', in outcome probabilities. The probability of exploring P(Explore) is the total number of choices made for the option with the lower expected value divided by the number of choices made in the session.

Sessions 1, 2 and 3; 27.5±0.8, 23.6±0.6, 24.8±0.9) but no main effects of DBS state (F(1,34) = 0.9, p=0.34) or interaction between DBS state and performance in a session of the task (F(2,34) = 1.03, p=0.30) see Figure 3A&B. The same analysis applied to the decision time (defined as the interval between the choices being presented and keypress) confirmed a significant main effect of session (F(2,34) = 5.21, p=0.005, DTs: Sessions 1, 2 and 3; 1.04±0.04, 1.14±0.05, 1.18±0.05 s) but no effect of DBS (F(1,34) = 0.01, p=0.92) or interaction between DBS and the decision times within a session of the task (F(2,34) = 1.27, p0.27). This analysis confirmed that switching their DBS stimulator to the OFF state did not affect task performance by deteriorating the patient's symptom control. Appling the same analysis, with group (HC or patients) and task session as fixed factors, the number of rewards obtained in each session by the patients was no different compared with the HC group (HC average rewards: Sessions 1, 2 and 3; 24.7±1.0, 22.7±0.8,28.1 ± 0.8; F(1,70) = 0.01, p=0.91) but the decision times in the HC group were on average faster compared to patients HC DTs: Sessions 1, 2 and 3 1.06±0.02, 1.03±0.01,0.95 ± 0.01 s main effect of group (F(1,70) = 5.36, p=0.02).

Our hypothesis was that DBS induced modulation of GPI would enhance exploratory choices. We defined the probability of making an exploratory choice, 'P(Explore)', as the proportion of choices made in a session where the patient chose the low-value option as in an e-greedy or softmax choice rule (*Sutton and Barto, 2018*). This categorisation relies on estimates of the expected values of the available two options for low-value, exploratory choices to be defined. To identify these, we fitted the patients' choices and DT's using the RLDDM (*Pedersen et al., 2017*). This model represents a fusion of two traditionally separate models which combine the process of iterative updating the value of expectations (Q-value *Figure 2A*) using the delta learning rule (*Rescorla, 1972*) with a choice rule (substituted for the traditional softmax) based on the sequential sampling mechanism of the Drift Diffusion Model (*Ratcliff, 1978*).

The RLDDM model (*Figure 2A*) includes four main parameters of interest (free parameters). First, the drift rate, *v*, reflects the average speed with which the decision process approaches the response boundaries *a* (which take on positive and negative values to represent the two choices in the task).

Because the *drift rate* on any one trial is proportionate to the difference in the expected value of the two choices (*Figure 2A* $Q_U$-$Q_L$) the scaling parameter, m, determines the extent to which the diffusion process is weighed by the difference in values of the two options. It is analogous to, and closely related to the beta parameter (inverse temperature function) which governs the explore-exploit trade-off in a softmax choice rule (*Pedersen et al., 2017*). The expected values for the choices are in turn derived from the delta learning rule $\alpha$ *(R-Q) with either a single learning rate, $\alpha$, or a model variant with separate learning rates for positive, $\alpha_+$ and negative $\alpha_-$ prediction errors. The nondecision time parameter, *t*, captures the time taken by stimulus encoding and motor processes. The model fit, as measured by the Deviance Information Criterion (DIC) (*Spiegelhalter et al., 2002*) indicated that the RLDDM with separate learning rates provided a better fit to the data compared to the model with a single learning rate (Patients: Dual learning rate DIC = 4344.98; single learning rate DIC = 4436.27; HC: Dual learning rate DIC = 3053.82; single learning rate DIC = 3056.00). Accordingly, all subsequent results reported are obtained using the dual learning rate model.

With the RLDDM model fitted to the choices separately for the ON and OFF-DBS states, we estimated the P(Explore) values for each session in the task (*Figure 2D*). A fixed effects analysis of P(Explore), was performed using a two-way repeated measures ANOVA with stimulation state (ON-DBS versus OFF-DBS) and task session as fixed factors. This demonstrated main effects of session (F(2,34) = 3.14, p=0.04) and DBS state (F(1,34) = 4.64, p=0.03) but no interaction between DBS state and the session of the task on the degree of exploration F(2,34) = 0.77, p=0.46. In the OFF-DBS state the average P(Explore) across the sessions was 0.13±0.03 which increased to 0.2±0.04 when the patients performed the task with the DBS stimulator switched ON (*Figure 3*). In *Figure 3*, the average P(Explore) values and reward performance for each session from 50 synthetic data sets is overlaid with the same values derived experimentally. These simulated choices were generated from RLDDM using the individual parameter estimates for each subject in the ON and OFF-DBS states from the tasks. This generated data was able to reproduce both the enhancing effect of DBS on P(Explore) and the preservation of reward learning performance in the ON and OFF-DBS states.

This analysis across the three experimental sessions in each task block may have averaged out more nuanced effects of DBS on exploration related to the reversal switch in contingencies, midway through session 2. To address this, we re-analysed the P(Explore) across twelve 10-trial bins (*Figure 3—figure supplement 1A*). A fixed effects analysis of P(Explore) was performed using a two-way repeated measures ANOVA with stimulation state (ON-DBS versus OFF-DBS) and task bin as fixed factors. This demonstrated main effect of bin (F(11, 187)=5.26, p<0.001), consistent with increased P(Explore) as an effect of contingency reversal and enhanced P(Explore) values in the ON-DBS state (F(1,187) = 9.84, p=0.001). Despite this, there was no interaction between DBS state and a specific bin of the task on the degree of exploration (F(11,187) = 0.51, p=0.89). In view of the marked increase in P(Explore) in the post-reversal bins 7–12 (*Figure 3—figure supplement 1A*), we also analysed the binned P(Explore) values with a fixed effect of reversal, allowing comparison of the P(Explore) values within pre-reversal bins (1-6) with those post-reversal (7-12). This demonstrated a main effect of reversal on the P(Explore) values (F(1,187) = 27.4, p<0.001), a main effect of DBS state (F(1,187) = 9.35, p=0.002), but no interaction between DBS state and reversal (F(1,187) = 2.54, p=0.11). The mean P(Explore) value in each 10 trial bin in the ON-DBS state was 0.2±0.04 and 0.14±0.05 in the OFF-DBS condition.

We defined the probability of a reward, P(Reward), as the number of rewards obtained in a 10-trial bin divided by the number of responses, to assess the performance in the task on a finer time scale. This was used rather than the absolute number of rewards (as used in the analysis of session performance above) as it was a more accurate measure of performance in a small trial bin as missed trials were accounted for. Using P(Reward) for each subject at each bin, we found no main effect of DBS (F(1,187) = 1.88, p=0.17), or interaction between the DBS and the task bin (F(11,187) = 0.74, p=0.7). Again, this analysis confirmed that enhanced exploratory choices in the ON-DBS state were not associated with degradation in the ability of the patients to aquire rewards during the task (mean P(Reward) values ON-DBS were 0.63±0.04 and 0.65±0.04 OFF-DBS; *Figure 3—figure supplement 1B*).

To place the effects of DBS in our patients into the context of known abnormal reinforcement learning behaviour in isolated dystonia (*Gilbertson et al., 2019*; *Gilbertson et al., 2020*), we compared the binned P(Explore) values of the HC group, to those of the patients in the OFF-DBS condition (*Figure 3—figure supplement 2*). The probability of the HC group making an exploratory choice was greater than the patients in the OFF-DBS condition (mean HC P(Explore)=0.19 ± 0.05),

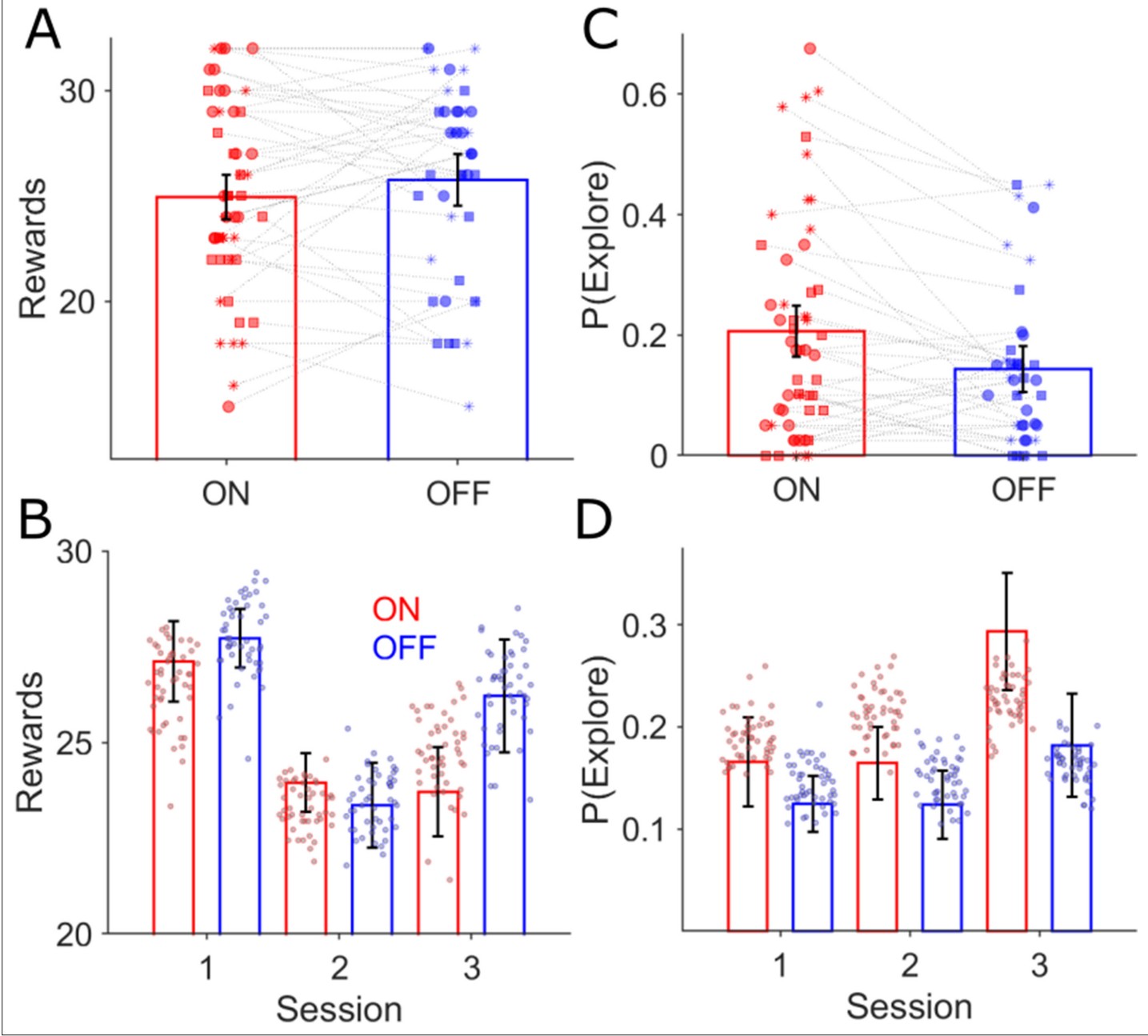

**Figure 3.** Behavioural effects of GPI DBS on task performance. There was no difference in the mean rewards (ANOVA, p>0.05) won across the three task sessions for the ON (red bars, n = 18) and OFF-DBS (blue, n=14) conditions (**A**) and (**B**) mean ± s.e.m. Each symbol represents a subject's number of rewards won for that session, with sessions one, two and three represented by the circle, square and asterix symbols. During ON-DBS testing the probability of exploring the lower value choice was significantly greater (ANOVA, p<0.05) (**C**). Behavioural performance is plotted in (**B**) (number of rewards) and (**D**) (probability of exploring the lower value choice) for both DBS conditions with the superimposed scatter plots of the 50 simulated experiments generated using the RLDDM fitted to the ON and OFF experimental choices.

The online version of this article includes the following figure supplement(s) for figure 3:

**Figure supplement 1.** Behavioural effects of GPI DBS on task performance peri- contingency reversal.

**Figure supplement 2.** Behavioural performance in Healthy Controls (HC) compared to patients in the DBS-OFF condition.

confirmed by a main effect of group (F(1,341) = 7.34, p=0.005). By contrast, there was no difference in the HC P(Explore) values when these where compared to the ON-DBS (F(1,385) = 1.1, p=0.3) and no effect of dystonia on the overall performance of the task, as P(reward) values in the HC group were the similar (HC; mean P(Reward)=0.63 ± 0.03) to those of the patients in the OFF-DBS condition (F(1,341)

= 2.48,=0.2). Overall, this analysis supported that GPI-DBS enhances exploratory choices in dystonia patients from a disease related baseline level of exploitation that is higher than age matched controls.

Finally, to ensure that the order of the two reversal tasks performed OFF and ON-DBS did not contribute in any way to this result, we ran a fixed effects analysis of P(Explore) across the three sessions with stimulation state (ON-DBS versus OFF-DBS) and task order as fixed factors. No main effect of task order on P(Explore) was evident from this analysis F(1,34) = 0.02, p=0.87, therefore, the enhanced exploration ON-DBS was unrelated to whether the first of the two reversal tasks were performed with DBS stimulator switched ON or OFF.

## Cortical functional connectivity correlates of the effect of DBS on exploration

When the DBS enhancing effects on exploratory choices on an individual subject level was investigated, this varied from a maximum within session increase of P(Explore) of 0.34, equating to 14 additional exploratory choices out of 40 in one subject, to no influence of DBS on exploration in any session in another. We leveraged the known influence of DBS on brain networks (*Horn et al., 2017*) to test the hypothesis that this variance could be explained by individual differences in connectivity of the DBS electrode with cortical regions which correlated strongly with exploratory decisions.

The connectivity analysis was performed based on established methodology previously introduced to assess brain networks affected by distributed lesions or stimulation effects in neurological and psychiatric conditions (*Boes et al., 2015*; *Corp et al., 2019*; *de Almeida Marcelino et al., 2019*).

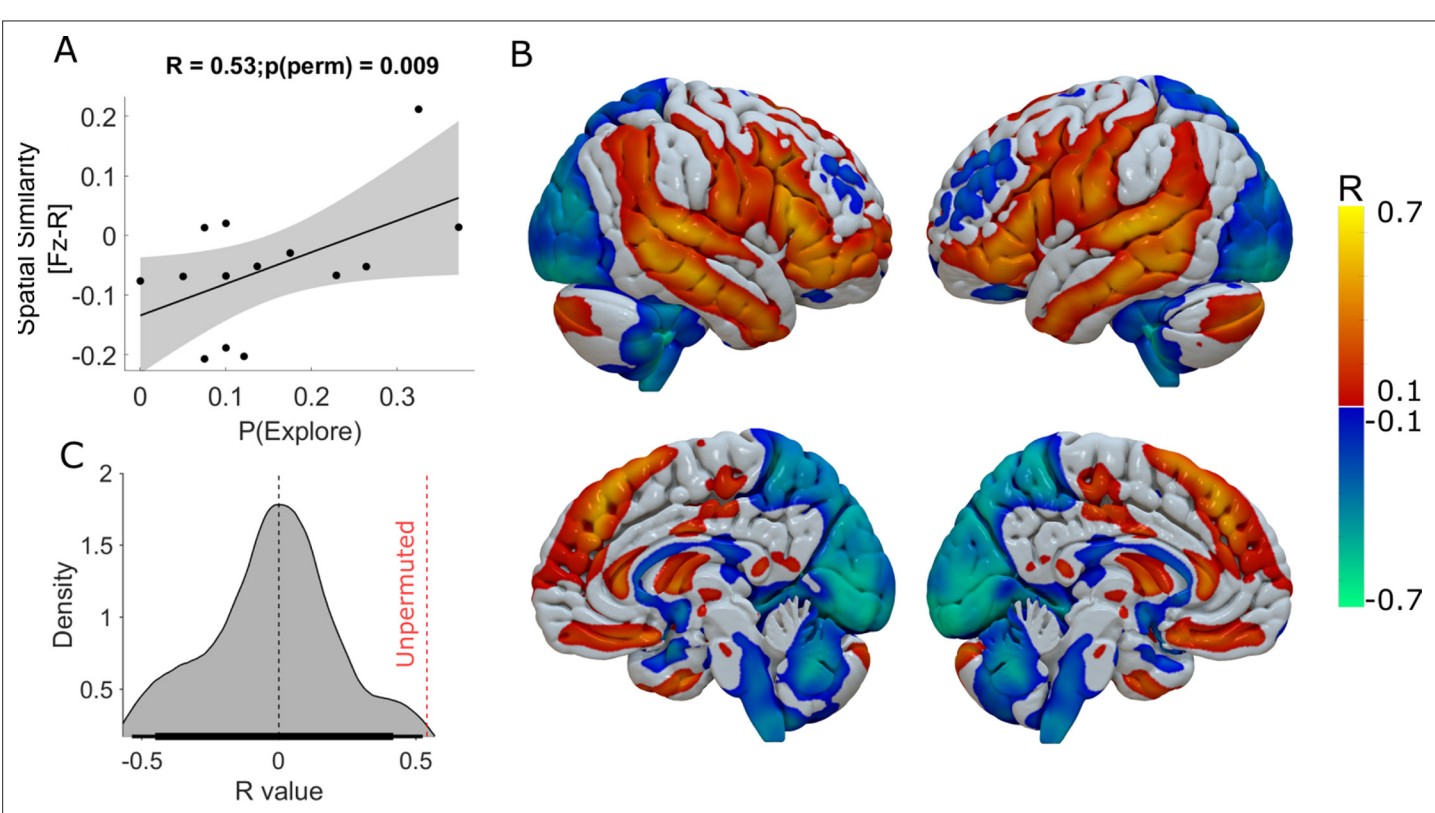

**Figure 4.** Functional connectivity of DBS-Induced exploration. The whole-brain voxel-wise R-Map demonstrates the optimal functional connectivity profile for DBS-induced enhancement of exploration (**B**). In this analysis, the maximal increase in DBS induced exploration (ON-OFF DBS) in one of the three experimental sessions was used as the regessor. Warm colours show voxels where functional connectivity to the DBS stimulation volumes was associated with greater exploration. Cool colours indicate voxels where functional connectivity to the DBS stimulation volumes was associated with lesser exploration. The more the individual functional connectivity profile matched the 'optimal' R-Map, the greater was the DBS-induced exploration (**A**) ($R^2$=0.28, p=0.04). In (**C**) we plot the R value distribution derived from 1000 repermuted correlations between the enhancing effect DBS and the R-map. The probablity of seeing the same correlation by chance was p<0.01 (represented by the red dashed vertical line).

The online version of this article includes the following figure supplement(s) for figure 4:

**Figure supplement 1.** Functional connectivity of DBS-induced exploration.

Using the stimulation volumes of the DBS electrodes as seed regions to a group connectome derived from resting state functional connectivity of healthy volunteers (*Yeo et al., 2011*), we derived whole-brain functional connectivity maps for each patient. Then, the correlation between functional connectivity and DBS-induced change in exploratory behaviour was calculated voxel-wise. The R-Map in *Figure 4B* depicts thresholded correlations between functional connectivity from DBS stimulation volumes to all brain voxels and DBS-induced exploration.

We used the difference in P(Explore) values ON minus OFF-DBS as the behavioural parameter for this analysis to normalise for between subject variation in the baseline level of exploration and to identify the super-added effect of DBS on this behaviour. For each subject, we chose the experimental session with the largest increase in DBS induced exploration. This aimed to minimise the effect of averaging this behaviour across the task as a whole, whilst allowing for individual differences in the effect of DBS across different sessions of the task not evident at a group level from the analysis of P(Explore). Spatial correlation of individual patients' DBS connectivity profiles with this R-Map significantly explained variance in DBS-induced exploration (*Figure 4A*) n=14 $R^2$=0.28 p=0.04, Permutation test: *R*=0.53, p=0.009. *Supplementary file 2* includes the details of the peak R-map values and voxel co-ordinates of the cortical regions whose connectivity with the DBS electrode predicted increasing exploration. To ensure that choosing the maximum within session difference in P(Explore) for this R-map did not arbitrarily identify network effects of pallidal neuromodulation, we performed an additional R-map analysis, averaging across the whole task and subtracting the difference in P(Explore) values in the ON minus OFF- DBS conditions (*Figure 4—figure supplement 1*). This revealed cortical connectivity patterns with similar topography. This R-map had a poorer linear predictive value of P(Explore), $R^2$=0.17, *P*=0.12 but met the permutation threshold (Permutation test: *R*=0.41, p=0.004) suggesting this was unlikely to occur by chance.

## Effects of DBS on decision choices revealed by influence on RLDDM parameters: model validation and parameter recovery

We performed posterior predictive checks to test whether the dual learning rate RLDDM was a good model of the experimental data. First, we compared the observed data, in our case the performance in the reversal task and the decision time distributions, to simulated data generated by the model. The learning curves (which illustrate the probability of choosing the highest value choice) are illustrated in *Figure 5A*. These represent the model's generative choices for fifty simulated 'experiments' with the RLDDM parameters estimated for each individual subject in both the ON and OFF-DBS conditions. The good model fit is indicated by the overlap between the generated data performance (shaded) with the observed choices (solid line) for both conditions. Furthermore, because the RLDDM includes a prediction of the decision time (DT) at which the choice was made, we also overlaid the observed DT histograms with the corresponding density for 50 generated simulations (*Figure 5C*). The close overlap between the observed and generated data supports that the RLDDM captured the salient features of the patients' choices and decision times whilst observing the heightened exploratory choice tendency in the ON-DBS state, as the same generative data was used for the overlaid P(Explore) simulated data in *Figure 3D*. As an additional check, we performed parameter recovery by re-fitting the first five generated data sets, plotting the model parameter estimates for the observed experimental data against the estimates derived from generated choices (*Figure 6D*). For all parameters, we found significant correlations between the parameters used to generate the behaviour and the recovered parameters obtained from re-fitting the model to the generated behaviour.

We estimated group and individual parameter values for the two learning rates ($\alpha_+$ , $\alpha_-$), the boundary separation parameter (*a*), and the drift rate scaling parameter, (*m*) as dependent variables of DBS state (ON or OFF). The non-decision time, *t*,(seconds), was estimated for each subject but was assumed not to be influenced by DBS state (as DBS did not appear to affect the absolute DT in the behavioural analysis above). Consistent with the finding of slower DTs in the patients compared to HC, the estimate for *t* in the patients was correspondingly longer (Patients = 0.57) [ 95% HDI 0.46, 0.69], HC = 0.42 [0.36 0.50] a difference in the posterior difference of the means which was statistically significant $M_{diff}$ = 0.13 [0.001, 0.27].

Group level parameters of the within-subject effects of DBS were used to assess how pallidal neuromodulation influenced exploratory choices. We found no statistically significant differences between the two DBS states when the dual learning rate model was fitted to all trials across the

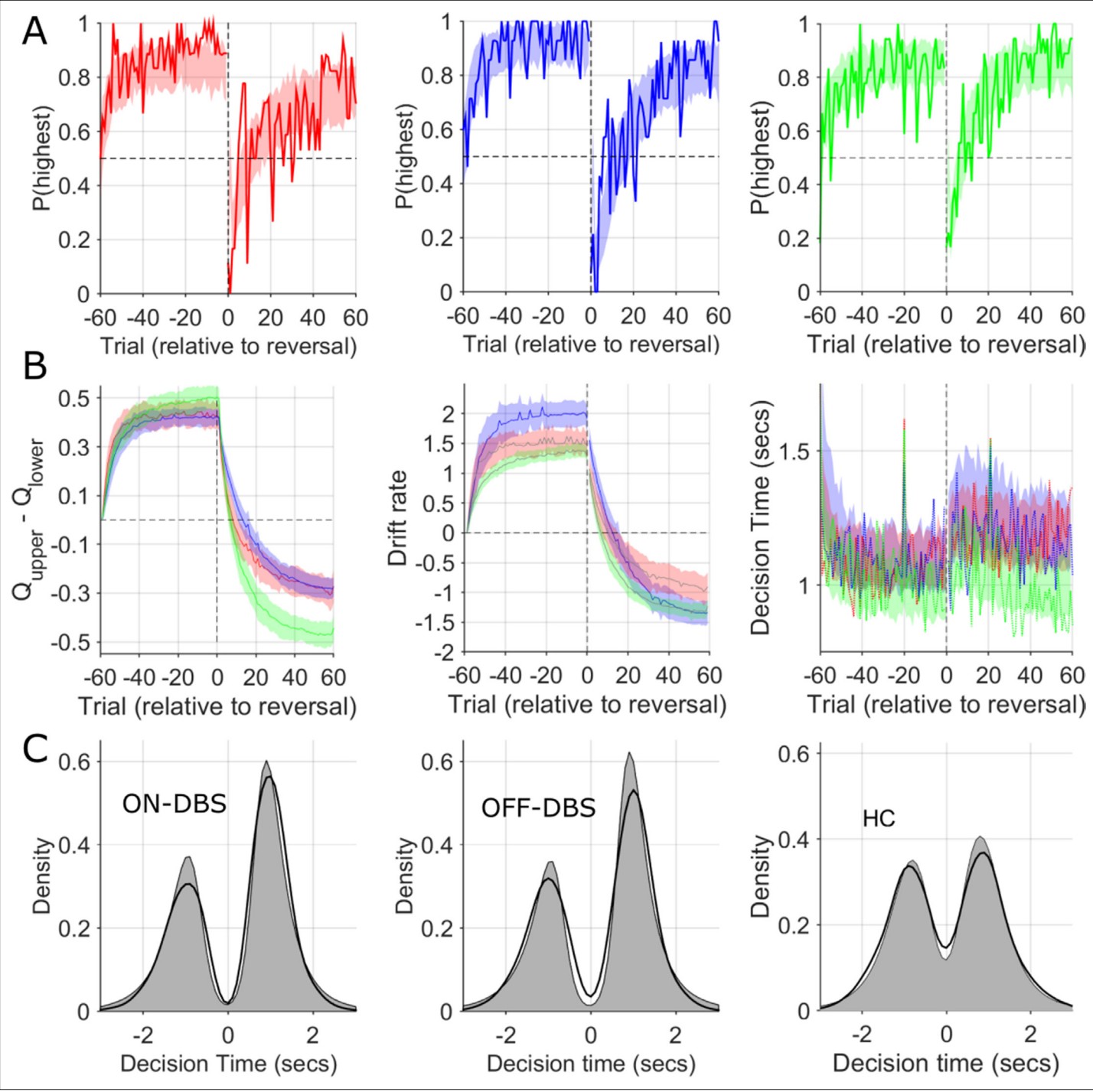

**Figure 5.** Observed and simulated RLDDM data. (**A**) The probability of choosing the highest value stimulus in the ON-DBS state is represented by the solid red line. Overlaid shaded area represents 95% confidence limits of simulated choices of the RLDDM (averaged across 50 simulated experiments). The same analysis for the OFF-DBS state is plotted in blue in the middle panel and for HC in green. This supported the interpretation of the model being a good fit to the experimental data due to the strong overlap between the synthetic data choices and those observed for both DBS conditions and in both HC and patient groups. (**B**) The difference in the expected value (**Q**) of the two choices in the RLDDM (mean ± 95% confidence limits) are represented by the solid blue (OFF-DBS), red (ON-DBS) and green (HC) lines with the shaded overlay showing the confidence limits estimated from the synthetic (simulated) data. The middle panel in (**B**) illustrates the trial-to-trial variation in the mean drift rate, *v*, across the simulations in all conditions and groups. This demonstrates a reduction in the ON-DBS drift rate consistent with the group level effect of DBS on reducing the drift rate scaling parameter, *m*, (see *Figure 6—figure supplement 1* and *Supplementary file 2*) and the lower drift rate in the HC group. The mean experimental trial-to-trial variation in DT in each DBS condition and the HC group are plotted as a solid line with the simulated model DT's overlaid. The model captures

*Figure 5 continued on next page*

*Figure 5 continued*

both the within task cost of the contingency reversal on DT and the faster DT in the HC group through the task. In (**C**), thicker black lines display the observed decision time distribution across all patients and sessions in the ON-DBS (red) and OFF-DBS (blue) conditions and HC groups, with the simulated DT from the RLDDM represented by the grey shaded regions. The decision time (DT) for choices which have an initially lower value are shown as negative. The reliability of the RLDDM in capturing the decision mechanisms in the task are supported by the overlap in observed and simulated DT distributions.

reversal learning task (*Figure 6* & *Table 1*). This model was therefore able to adequately capture the patient's exploratory decisions in the task and the effect of DBS (*Figure 3B&D*, *Figure 5*, *Figure 6D*) but provided no mechanistic insight into how pallidal neuromodulation drove an increase in these choices. Given the strong influence of the contingency reversal mid-way through session 2 of the task on exploration, (*Figure 3—figure supplement 1*), we refitted the model separately to the pre-(1-60) and post-reversal trials (61-120). The posterior distributions of the four parameters and their posterior differences (ON minus OFF-DBS) are illustrated in *Table 1—source data 1* and *Figure 6—figure supplement 1*. We found a similar pattern of posterior directional effects of DBS when fitting to the whole task or separately to pre- and post-reversal. DBS led to increases in both learning rates, decreases in the boundary separation parameter (*a*) and drift rate scaling parameter (*m*). However, analysing the pre- and post-reversal trials independently, we detected the only statistically significant group level difference in the drift rate scaling parameter, *m*, in the post-reversal phase of the task; $M_{diff}$ = - 1.71 (95% HDI [-3.5,–0.18]). This reduction in the drift rate scaling parameter *m* ON-DBS is consistent with enhanced exploration, as this parameter amplifies the influence of the difference in the expected value of the two choices ($Q_u$ - $Q_L$), by driving the diffusion process to the decision boundary of the higher value option (*Figure 2A*).

Comparing the RLDDM parameter estimates between the HC and patients in the OFF-DBS condition, the estimated value of, *m*, was also reduced in the HC group relative to the patients in the DBS-OFF state *Figure 6—figure supplement 2*, *Table 1—source data 2*; $M_{diff}$ = - 1.71 (95% HDI [-3.5,–0.18]). This explains the higher proportion of exploratory choices by HC's in the task for the same reason that in the ON-DBS state exploration is heightened – this parameter proportionately scales the influence of the difference in values between the two options in on the eventual choice of decision. Patients with dystonia have previously been shown to have impaired learning from negative feedback (; *Gilbertson et al., 2020*). Consistent with these previous studies we also found statistically significant difference in the negative learning parameter, $\alpha_-$, with a higher value HC group compared to the patients in the OFF-DBS condition $M_{diff}$ = 1.32 (95% HDI [0.26,2.5]).

This comparison of RLDDM parameter estimates between the HC and OFF-DBS groups confirmed that greater exploratory choice tendency in HC's could be explained by a lower influence of the difference in expected value of two choices. In both analyses, lower levels of exploration in HC's compared to patients with dystonia and enhanced exploration by GPI neuromodulation in the patients ON-DBS, were most likely explained in the RLDDM by relative reductions in the drift rate scaling parameter (*m*). In turn, this meant that in both the HC group and in the patients in the ON-DBS condition, the decision to choose one of the two options in each trial was much less influenced by encoding of their value. Accordingly, choices were influenced proportionately more by noise intrinsic to the decision process, resulting in greater proportion of random, exploratory, (and accordingly less greedy, exploitative) choices. This was in turn reflected by a higher level of P(Explore) values both ON-DBS and in HC's relative to the OFF-DBS condition in our behavioural analysis above.

## Discussion

In this study, we examined the role of the human pallidum (Globus Pallidus Interna) in mediating the trade-off in explore-exploit decision-making. Our two predictions were confirmed: (1) Neuromodulation by deep brain stimulation of the GPI increased the likelihood of patients exploring the lower value choice in a two-armed probabilistic reversal learning task; (2) DBS-induced enhanced exploration correlated with the functional connectivity of the stimulation volume in the GPI to a distributed brain network including frontal cortical regions identified previously in functional imaging studies of explore-exploit decision-making. Furthermore, a recently proposed reinforcement learning model successfully predicted the behaviour, enabling a more mechanistic interpretation of experimental results.

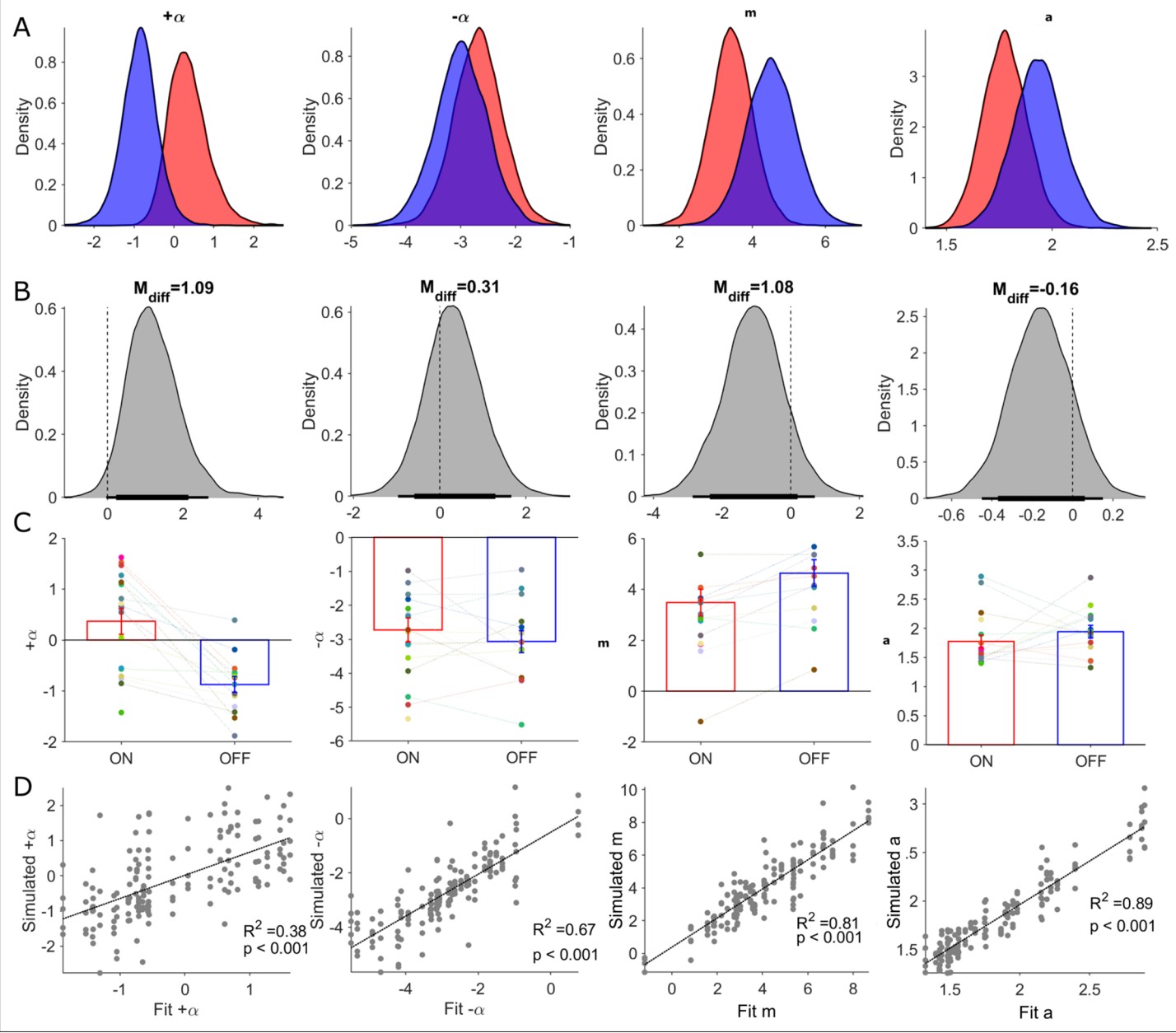

**Figure 6.** Posterior distributions of RLDDM parameters and parameter recovery. Columns in this figure display results for each of the estimated parameters in the RLDDM from left to right, including positive learning rate ($\alpha_+$), negative learning rate ($\alpha_-$), drift rate scaling parameter (**m**), boundary separation parameter (**a**). Rows, from top to bottom, correspond to (**A**) posterior distributions ON-DBS (red), OFF-DBS (blue). (**B**) Posterior distributions of differences ON-DBS versus OFF-DBS. Thick and thin horizontal bars below the distributions represent the 85% and 95% highest density intervals, respectively. (**C**) Mean ± S.E.M. parameter estimate ON-DBS (red) and OFF-DBS (blue). Each individual's parameter estimate represented by a different colour in the scatter plot for each parameter. (**D**) Results of parameter recovery analysis, plotting the estimated parameter values for the observed data against the parameter values re-estimated from simulated data. Significant (p<0.001) linear correlations were observed for all four parameters supporting successful recovery and validation of applying the RLDDM model of decision-making.

The online version of this article includes the following figure supplement(s) for figure 6:

**Figure supplement 1.** Posterior distributions of RLDDM parameters and parameter recovery – separately fitted to pre- and post- reversal trials.

**Figure supplement 2.** Posterior distributions of RLDDM parameters from Healthy Controls (HC) and parameter recovery.

**Table 1.** Summary of posterior distributions (RLDDM model fit to whole task).

| Parameter | ON-DBS | | | OFF-DBS | | | Contrast | | |
| --- | --- | --- | --- | --- | --- | --- | --- | --- | --- |
| | *mean* | HDI | | *mean* | HDI | | *mean* | HDI | |
| Boundary separation | 1.78 | 1.56 | 1.98 | 1.96 | 1.7 | 2.18 | –0.16 | –0.45 | 0.15 |
| Drift rate scaling | 3.4 | 2.26 | 4.54 | 4.51 | 3.20 | 5.88 | –1.08 | –2.86 | 0.71 |
| Learning rate + | 0.22 | –0.55 | 1.37 | –0.85 | –1.79 | 0.01 | 1.09 | –0.03 | 2.68 |
| Learning rate - | –2.65 | –3.58 | –1.8 | –2.97 | –3.97 | –2.05 | 0.31 | –0.97 | 1.66 |

Estimated means (m) for ON and OFF-DBS conditions as well as contrast for ON-OFF. HDI values are for 95% highest density Interval.

The online version of this article includes the following source data for table 1:

**Source data 1.** Summary of Posterior distributions (RLDDM model fitted separately to pre- and post-reversal trials).

**Source data 2.** Summary of Posterior distributions (RLDDM model fitted to whole task).

## DBS effects on GPI and their contribution to explore-exploit choice behaviour

The exploration – exploitation 'dilemma' is considered one of the fundamental challenges of adaptive control and behaviour (*Mehlhorn et al., 2015*; *Cohen et al., 2007*). We took advantage of the opportunity leveraged by DBS to address the paucity of experimental evidence (*Sheth et al., 2011*) relative to theoretical support for the BG role in explore-exploit choice arbitration (*Gilbertson and Steele, 2021*; *Humphries et al., 2012*; *Chakravarthy et al., 2010*).

Patients and non-human primates with lesions of the GPI exhibit an inability to learn novel stimulus response contingencies and random choice behaviour which prevents the acquisition of choice preference or implicit learning of sequences (*Piron et al., 2016*; *Obeso et al., 2009*). We did not observe levels of performance degradation in our patients that would be comparable to complete lesioning of pallidal output.

Neurophysiological data support a net inhibitory effect of DBS on GPI neuron firing rates without complete firing suppression at stimulation intensities comparable to those used in our patients (*Bar-Gad et al., 2004*). We suggest that the most likely neuromodulatory mechanism of DBS on enhanced exploration is a partial inhibition of GPI firing and accordingly electrically lesioning the basal ganglia's output (*Wu et al., 2001*; *Dostrovsky et al., 2000*; *McCairn and Turner, 2009*; *Boraud et al., 1996*).

Previous studies have confirmed that during acute changes in GPI-DBS (comparable to the ON-OFF conditions tested in our patients) a reduced inhibitory output induced by DBS in GPI leads to increased excitability of the motor cortex (*Tisch et al., 2007*; *Kühn et al., 2003*). Motor cortex activity correlates negatively with decision value during explore-exploit decisions (*Tomov et al., 2020*). This means that within a sequential sampling decision mechanism, increased motor cortex activity correlates with small differences in the value of the two choices and corresponding higher decision uncertainty and choice exploration. This was indeed the effect predicted by the RLDDM in the ON-DBS state. A reduction in the drift-rate scaling parameter (m) has a corresponding effect on decision value as this parameter multiplies the difference in expected value in the two choices (*Pedersen et al., 2017*). The exploratory influence of pallidal DBS could therefore be explained by thalamocortical disinhibition, increased motor cortex excitability and its activity inversely encoding decision value.

Alternatively, rather than an influence on a downstream cortical decision mechanism, GPI-DBS may have influenced implementation of this process *within* the BG circuit. The simulations of *Dunovan et al., 2019* propose that activity in the direct (striato-nigral) and indirect (striato-pallidal) BG pathways differentially encode the rate of evidence accumulation and the amount of information needed to reach decision threshold. We found that GPI-DBS reduced the drift rate scaling parameter during the post-reversal phase of the task. As this parameter determines the rate of evidence accumulation, our results would be consistent with this predicted function of the direct pathway being neuromodulated by pallidal DBS. However, because the GPI sits anatomically at the point of convergence of both direct and indirect pathways, we cannot distinguish this mechanism from a downstream influence on thalamo-cortical excitability. Future studies examining neuromodulation of a basal ganglia nucleus

that lies within the indirect pathway, such as the subthalamic nucleus (STN), could be used to delineate this mechanism further, as this might be predicted to selectively influence the RLDDM boundary separation parameter which determines the decision threshold.

## The basal ganglia's role in reinforcement learning (information seeking vs. uncertainty)

During the early stages of learning, firing rates of GPI neurons show transient inhibitory pauses in firing rate which encode exploratory choices. These are replaced with higher firing rates as choices become habitual and stimulus response mapping is obtained (*Sheth et al., 2011*). These findings are consistent with physiological changes in the direct and indirect pathways during different stages of learning. (*Yin et al., 2009*) identified increased excitability in the striato-nigral (D1R-expressing) direct pathway early in learning with corresponding inhibition of the GPI. With subsequent consolidation of learning and habituation of the stimulus response relationship, D2R dependent increases in striato-pallidal excitability would dis-inhibit the GPI (*Kravitz et al., 2010*), increasing its excitability and promoting exploitation. Recent data from *Lee and Sabatini, 2021* would however contradict this interpretation as they found exploration increased in optogenetically stimulated indirect pathway striato-pallidal neurons. This discrepancy might be explained by careful distinction between basal ganglia contributions to actively promoting exploratory choices during learning, versus active inhibition of choices from negative outcomes, that may be more relevant to choice extinction (*Tecuapetla et al., 2016*) and shifting (*Kravitz et al., 2012*).

An alternative explanation (to the idea of active exploratory information seeking) is that enhanced non-greedy choices ON-DBS represent increased random variability in the update of action values due to computational noise inherent to the learning process (*Findling and Wyart, 2021*). Examining the effects of GPI-DBS in a non-stationary 'restless' bandit task could delineate the contribution of decision noise as these models that take noise into account of the decision process have been successfully fitted to such tasks (*Findling et al., 2019*).

## The basal ganglia's role in reinforcement learning (learning from previous outcomes)

We found no effect of DBS on the learning rate to either positive or negative outcomes in the task. This is surprising given evidence from Local Field Potential (LFP) recordings from GPI-DBS electrodes which demonstrate that these encode choice outcome (*Eisinger et al., 2022*). Reward Prediction Error (RPE) encoding in LFP's from GPI electrodes have also been identified and are most marked during exploratory decisions and absent when choices are exploitative (*Schroll et al., 2015*). Pallidal error-related activity has been identified from LFP recordings. This proceeds the cortical error-related negativity (ERN), which is, in turn, closely related to the negative prediction error signal (*Herrojo Ruiz et al., 2014*; *Holroyd et al., 2003*). A plausible a priori mechanism for increased exploration would be DBS induced blunting of these learning signals. The absence of any effect of DBS on the RLDDM learning rate parameters results would suggest that outcome signalling including RPE-like and ERN signals in the GPI are relatively unaffected by DBS. Therefore, rather than modify intrinsic encoding of RPE-like signals or through distal connections with other brainstem structures, (*Hong and Hikosaka, 2008*) DBS enhanced exploration appears to be mediated by a neuromodulation of the decision process, consistent with theoretical accounts proposing the basal ganglia's excitability as an arbitrator of action selection (*Gilbertson and Steele, 2021*; *Humphries et al., 2012*; *Dunovan et al., 2019*).An equally parsimonious explanation for the absence of any impact of DBS on the learning rate, is that by targeting of the (dorsal) motor segment of GPI, there was no effect on the excitability of the ventrally located GPI – lateral habenula pathway (*Hong and Hikosaka, 2008*) which facilitates negative prediction error signalling in the Ventral Tegmental Area (VTA).

## Integrating the results in a network perspective

The postero-lateral portion of the GPI targeted in the group of patients studied here is chosen in routine clinical practice because of its position within the sensorimotor loop of the basal ganglia (*Alexander et al., 1986*) to relieve motor symptoms of dystonia. It is noteworthy that despite evidence for anatomically segmented regions of the GPI (based on white matter connectivity *Draganski et al., 2008*) stimulating the sensorimotor pallidum in this study could alter decision-making and

therefore modify 'cognitive' function. At first glance, this result might appear incongruous with its established role in motor control. However, our functional connectivity analysis supports the conclusion that sensorimotor GPI function extends beyond that traditionally attributed to it on the basis of anatomical connections. The finding of a neuromodulatory influence of sensorimotor pallidum on decision-making is consistent with fMRI activation of pre-and post-central gyri, during exploratory choices (*Chakroun et al., 2020*) and the encoding of decision value by motor cortex (*Hare et al., 2011*).

Our connectivity analysis is noteworthy for the extent of the connectivity pattern across much of the brain. The largest within session effect of DBS was seen in patients where the electrode stimulation volume shared connectivity with brain regions which would be considered functionally relevant to our task (e.g. pre-frontal regions). Equally, this connectivity profile included regions with little to no recognised functional role in reward-reversal learning. Therefore, caution is required in interpreting the anatomical detail of this network given the limitations of using imaging data with no individualised anatomical specificity. On a more general level, this analysis provides support to the idea that discrete functions in local brain networks (in our case the GPI), are integrated into a broader scale whole-brain network. This leads to co-variation in neural activity between regions which are not conventionally attributed a functional role in a specific motor or sensory act (*Kauvar et al., 2020*) and might explain how DBS exerts influence on remote areas beyond the stimulation site (*Horn et al., 2017*). For instance, such as modulation of distributed networks through subthalamic DBS affects motor learning and risk-taking behaviour in Parkinson's disease patients (*de Almeida Marcelino et al., 2019*; *Irmen et al., 2019*).

## Clinical relevance - apathy

From a clinical perspective, these findings may be relevant to understanding the mechanisms which worsen apathy in Parkinson's disease (PD) following DBS targeting the basal ganglia (*Zoon et al., 2021*). Patients with PD-Apathy tend to preferentially choose high value outcomes over options with lower payouts (*Le Heron et al., 2018*). In the ON-DBS state, heightened exploratory choices observed in our data would lead to poorer encoding of the difference in value of decisions and incentivise only actions which lead to large differences in expected outcomes. This assumes that the mechanism of DBSinduced apathy is analogous to apathy acquired from lesions of the bilateral globus pallidus, where the basal ganglias output is nullified, and a decrease in the signal to noise ratio that encodes value to incentivise action has been proposed (*Levy and Dubois, 2006*).

## Study limitations

Patients with primary forms of dystonia who have not undergone DBS exhibit abnormal reinforcement learning (*Gilbertson et al., 2019*; *Arkadir et al., 2016*; *Gilbertson et al., 2020*). Comparison of decision making between our patients and a healthy control group replicated previously described RL abnormal bias towards explotative choices in these patients (*Gilbertson et al., 2019*; *Arkadir et al., 2016*; *Gilbertson et al., 2020*). It seems likely that the observed enhancement of exploration by pallidal neuromodulaiton of GPI is from a high level of exploitaiton that is intrinsic to an imbalanced basal ganglia circuit related to the disease process in dystonia patients. We therefore cannot discount the possibility that DBS enhanced exploration was a consequence of neuromodulating a pathologically biased decision making circuit. Given the previous experimental (*Sheth et al., 2011*) and theoretical (*Humphries et al., 2012*) support which motivated testing the hypothesis of this study, it seems likely that the same function is subserved by a physiologically intact basal ganglia circuit. Furthermore, allowing for the uniqueness of this clinical indication for DBS in this patient group, these interpretational limitations are offset by the infrequent opportunities to study the role of this brain region in human explore-exploit decision-making.

The use of normative connectomes instead of patient-specific data enhance the signal to noise ratio and image quality (*Horn et al., 2017*) but prevents individual quantification of connectivity strength in relation to the behavioural effect of DBS. Lastly, computational models account for a heuristic explanation of complex and dynamic neural networks and serve mainly as a theoretical support for the observed experimental results. The transferability of computationally inferred mechanisms remains a limitation of this study.

## Conclusions

A recent increase in interest in the explore-exploit dilemma has significantly advanced understanding of the functional neuroanatomy, with computationally demanding cortical algorithms resolving multiple-alternative decision problems (*Gershman, 2018*; *Schulz and Gershman, 2019*). The experimental results presented here demonstrate that a understanding of a unified account of the brain's approach to adaptive behavioural control benefits from inclusion of subcortical circuits. Future research should aim to delineate the differential contributions of subcortical and cortical circuits to explore-exploit decision-making as well as their interaction, leading to insights into disordered decision-making in psychiatric and neurologic conditions (*Addicott et al., 2017*).

# Materials and methods

## Participants

Nineteen patients with isolated dystonia (14 f, 59.79±1.93 years old; mean ± S.E.M) who had undergone Deep Brain Stimulation (DBS) surgery targeting the bilateral Globus Pallidus interna (GPI) were enrolled in the study. One participant was unable to complete more than ~50% of trials in the ON-DBS state and was excluded from the final analysis. Three of the remaining 18 patients were able to complete reversal-learning task in the ON-DBS state but were unable to tolerate testing in the OFF-DBS state. One patient's OFF-DBS testing file was inadvertently overwritten and was therefore not available for analysis. For more clinical details of the patients and specifications of the inserted electrodes please see *Supplementary file 1*.

Eighteen healthy controls (14 f, 57.31±1.13 years old) with no previous diagnosis of a neurological or psychiatric disorder were enrolled.

In accordance with the declaration of Helsinki, participants gave written informed consent to participate in the study, which was approved by the local ethics committee (Charité – Universitätsmedizin Berlin, EA1/179/20).

## Experimental task

We used a modified version of a two-choice reward reversal-learning task based on *Pessiglione et al., 2006*. The task was presented on a laptop screen using Psychtoolbox v3.0 (*Brainard and Vision, 1997*) running on MATLAB (R2019, The MathWorks, Natick, MA, USA). The laptop was positioned in front of the patient so that responses could be made on the keyboard. Participants were instructed to try to win as many 'vouchers' as possible. Printed screen text was in German. At the beginning of each trial, a fixation cross presented at the centre of the screen indicated that a new trial had begun. A pair of fractal images were presented and subjects were expected to indicate their choice by a keyboard button press. The order of the fractals was randomly assigned to either the right or left of the fixation cross. In the event that they did not make a response within 3 s the fixation cross was re-presented and a new trial began. A choice was highlighted for 0.5 s by a red circle around the chosen fractal. The choice screen was followed by feedback screen for a further 1 s. Feedback consisted of either – 'you win' or a neutral feedback condition with a screen with the words 'nothing'. Patients performed three sessions of 40 trials with 3–4 min breaks between sessions to improve task compliance. The visual stimuli were associated with a fixed probability of rewarding outcome of 80:20%. Reward contingencies were reversed after 60 trial presentations (i.e. the image associated with a 80% reward probability in the first 60 trials was associated with 20% reward probability in the last 60 trials and vice-versa). Participants were not informed about the existence of a contingency reversal. A short training session of 10 trials with a novel pair of fractals was performed before formal testing began to familiarise the patients with the task.

The initial DBS condition was randomised. Patients repeated the same task OFF-DBS in the opposite DBS state after a 20 min break. The fractal images differed between both task versions to avoid learning effects.

## Reinforcement learning drift diffusion model

The RLDDM model consists of a hybrid of the sequential sampling model choice rule based on the DDM with the drift rate, $v$ modified by the difference in the expected values of the two choices. The

DDM calculates the likelihood of the decision time (DT) of choice,, on trial, $t$ ,with the Weiner first-passage time (WFPT) distribution;

$$DT(x) \sim WFPT\left[a, T_{er}, z, v(t)\right],  \qquad (1)$$

The non-decision time and bias (starting point) of the diffusion process are represented by $T_{er}$ and $z$. The boundary separation parameter $a$ defines the point at which the diffusion process reaches the decision threshold and a response is estimated. The expected values, $Q_{upper}$ and $Q_{lower}$ of the choices represented by the upper and lower response boundaries, are estimated for each choice using the temporal difference learning rule (**Rescorla, 1972**).

$$Q_t = Q_{t-1} + \alpha * (R - Q_{t-1}),  \qquad (2)$$

where $\alpha$ is the learning rate and $R$ is the outcome of the choice on trial $t$. We fitted two variations of the RLDDM model, the first with a single learning rate, the second with dual learning rates, where the Q values were modified separately by a learning rate $\alpha^+$ that multiplied the positive prediction errors (i.e. outcomes that were better than expected) and by $\alpha^-$ , which multiplied negative prediction errors (when outcomes where worse than expected). The drift rate on each trial is then influenced by the difference in the expected values in the two choices by the drift rate scaling parameter $m$ , where,

$$v(t) = \left[Q_{upper}(t) - Q_{lower}(t)\right] * m.  \qquad (3)$$

Given our hypothesis and prior knowledge that the learning rate, scaling parameter and boundary separation parameters on explore-exploit choices (**Pedersen et al., 2017**), we estimated values of $T_{er}$ and $z$ with the a priori assumption of no effect of DBS and estimated values of $\alpha^+, \alpha^-$ , $m$ and $a$ as dependant variables for both on and OFF-DBS states.

Model parameter estimation was performed using the Hierarchical Bayesian framework implemented in Python (3.6.12) which uses the Hierarchical estimation of DDM (HDDM) module (**0.8.0;** https://hddm.readthedocs.io/en/latest/; **Wiecki et al., 2013**). All models were run with four chains with 7,000 burn-in samples and 15000 posterior samples each. Convergence between the Markov chain Monte Carlo (MCMC) chains was assessed using the Gelman Rubin statistic (**Gelman, 2013**). For all estimates including those for parameter recovery the Gelman Rubin values were all less than 1.1 indicating successful convergence.

Posterior distributions in the differences in the parameter estimates were estimated by subtracting the posterior estimates for each parameter in the ON and OFF-DBS states. The posterior means and highest density intervals (HDI) were calculated using the 'bayestestR' and 'tidybays' packages in R version 4.1.1 http://www.r-project.org (**Kay, 2019**; **Makowski et al., 2019**).

Mapping of the influence of the difference parameters on the exploratory choice preference in the RLDDM model was performed by simulating 20 synthetic data sets at each parameter combination and estimating the P(Explore) values. Parameter space mapping was performed for the positive learning rate, drift rate scaling and boundary separation parameters ($\alpha^+$ , $m$ and $a$). The parameter values for each simulation where centred around the OFF state group mean estimates with each combination of parameters derived from a range across of values relevant to the parameter space estimated from fitting the experimental data.

### Lead localisation and connectivity analysis

DBS Electrodes were localised for 14/19 of the patients whom both testing data was available ON and OFF-DBS using the software 'Lead-DBS' V2.5 (**Horn et al., 2019**) https://www.lead-dbs.org/ as described (**de Almeida Marcelino et al., 2019**; **Neumann et al., 2018**) and mapped in MNI template space. Stimulation volumes were modelled based on SimBio-FieldTrip pipeline incorporated in Lead DBS (**Bhatia et al., 2018**). Stimulation volume modelling was informed by DBS parameters used to control patients' symptoms during behavioural task (**Supplementary file 1**). Each patient's stimulation volumes were then used as seed regions to estimate individual functional connectivity profile maps using an openly available group connectome derived from resting state functional connectivity MRI images of 1000 neurologically healthy volunteers (**Holmes et al., 2015**). Then, voxel-wise correlations of individual connectivity to DBS-induced changes in the probability of exploring the lower value choice 'P(Explore)' were calculated and visualised as R-maps (**Figure 4B**) after thresholding R

(*Horn et al., 2017*; *de Almeida Marcelino et al., 2019*; *Neumann et al., 2018*). The DBS induced change in P(Explore) was defined as the maximum increase in P(Explore) ON-DBS minus P(Explore) OFF-DBS in one of the three sessions. The similarity between the 'ideal' connectivity profile of this R-map (*Figure 4B*) and the connectivity profile of each individual patient was mathematically assessed by calculating spatial correlation coefficients between the R-map and the individual non-behavioural connectivity maps (*Horn et al., 2017*; *AlFatly et al., 2019*). Finally, the predictive potential of the DBS connectivity maps for the increase in exploration was estimated by correlating the individuals increase in P(Explore) with DBS to the spatial correlation coefficient of each patient (*Figure 4A*). A permutation distribution was constructed from the same analysis performed 1000 times and the correlation coefficient re-estimated with the P(Explore) values for each subject randomly re-ordered.

### Generating a heat-map of stimulation volumes

This section describes a method to highlight the subcortical cluster stimulated by most of stimulation volumes in our cohort and its relation to GPi/GPe nuclei (extracted from the DISTAL atlas *Ewert et al., 2018*). This is important to explore the extent of stimulation and possible differential modulation of the basal ganglia nuclei in the vicinity of the stimulated area. The DISTAL atlas is a probabilistic atlas (which means it offers voxel-wise probability that a voxel is belonging to a specific structure). Therefore, we first thresholded bilateral GPi and GPe images to include only voxels with 50% probability of belonging to the GPi or the GPe and extracted binary masks of them. Later, stimulation volumes were extracted and saved as binary Nifty images as has been described in the main manuscript method section. We then overlapped all binary stimulation volumes (n=14) to extract a total stimulation volume representative of the full cohort. The latter has been explored and depicted as a 3D volume in relation to the GPi/GPe (*Supplementary file 1*) to visualise the extent of overlap with each of these structures. Next, we summed up the number of stimulation volumes contributing to each voxel in the total stimulation volume. This helped extracting an n-map (or heat-map) which can illustrate the frequency with which each voxel is being stimulated by stimulation volumes in the cohort. As a final step, we have calculated the extent of overlap between the heat-map and the GPi or GPe nuclei as the weighted sum of overlapping voxels. This means that we summed the values of the heat-map intersecting voxels with the binary mask of each of GPi or GPe nucleus. The latter method ensures that the voxel-wise frequency information would not be lost compared to simple binary sum of the overlap of total stimulation volume with GPi/GPe binary masks. All analyses mentioned in this section were performed in a grid of 0.5×0.5 × 0.5 mm resolution using bihemispheric information in respect of GPi/GPe or stimulation volumes (since all patients have been bilaterally stimulated).

### Behavioural data analysis

Prior to statistical analyses, trials without responses (errors of omission) were excluded ON-DBS mean 1.06±2.26 (range 0–8) trials, OFF-DBS 1.78±2.5 (range 0–7 trials). Accordingly, DT and task performance (number of rewards won in a session) were analyzed using ANOVA with the MATLAB function *anovan*. In case of significant interactions, post-hoc tests were conducted using paired samples t-tests. Normality assumptions were tested using Kolmogorov Smirnov tests (all p>0.05). All results are reported as mean values ± S.E.M.

## Acknowledgements

TPG was funded by a NRS Fellowship from the Chief Scientist Office (Scotland). Additional funding for this research was provided by the University of Dundee/NHS Tayside Movement disorders research Endowment Fund. AL de A Marcelino is a fellow in the BIH Charité Junior Clinician Scientist Program funded by Stiftung Charité.

## Additional information

#### Competing interests
Andrea A Kühn: has received from honoraria from Boston Scientific, Medtronic and Teva. The other authors declare that no competing interests exist.

## Funding

| Funder | Grant reference number | Author |
|--------|------------------------|--------|
| Chief Scientist Office | | Tom Gilbertson |
| NHS Tayside Movement disorders research Endowment Fund | | Tom Gilbertson |
| Stiftung Charité | | Ana Luisa de A Marcelino |

The funders had no role in study design, data collection and interpretation, or the decision to submit the work for publication.

## Author contributions

Ana Luisa de A Marcelino, Conceptualization, Formal analysis, Investigation, Writing – original draft, Writing – review and editing; Owen Gray, Formal analysis, Methodology, Writing – review and editing; Bassam Al-Fatly, Formal analysis, Visualization, Methodology, Writing – review and editing; William Gilmour, Data curation, Investigation, Writing – review and editing; J Douglas Steele, Supervision, Investigation, Methodology, Writing – review and editing; Andrea A Kühn, Conceptualization, Resources, Supervision, Writing – review and editing; Tom Gilbertson, Conceptualization, Formal analysis, Supervision, Funding acquisition, Investigation, Visualization, Methodology, Writing – original draft, Writing – review and editing

## Author ORCIDs

Ana Luisa de A Marcelino http://orcid.org/0000-0002-3291-7222
Bassam Al-Fatly http://orcid.org/0000-0003-0067-6177
Tom Gilbertson http://orcid.org/0000-0002-9866-1565

## Ethics

Human subjects: The which was approved by the local ethics committee (Charité - Universitätsmedizin Berlin, EA1/179/20).

## Decision letter and Author response

Decision letter https://doi.org/10.7554/eLife.79642.sa1
Author response https://doi.org/10.7554/eLife.79642.sa2

# Additional files

## Supplementary files

• MDAR checklist

• Supplementary file 1. Clinical Demographics. TWSTRS: Toronto Western Spasmodic Torticollis Rating Scale; BDI: Beck's Depression Inventory; F: Female; M: Male; S.E.M: standard error of mean; L: left; R: right. *These patients only performed the task in one stimulation condition.

• Supplementary file 2. Summary of Connectivity (R-map) analysis. ACC, anterior cingulate cortex; BA, Brodmann area; CBM, cerebellum; IFG, inferior frontal gyrus, ins., insula; ITG, inferior temporal gyrus; MCC, midcingulate cortex; MedFG, medial frontal gyrus; MFG, middle frontal gyrus; MTG, middle temporal gyrus; OG, orbital gyrus; OL, occipital lobule; PCC, posterior cingulate cortex; PreCG, precentral gyrus; Prec, precuneus; PostCG, postcentral gyrus; SFG, superior frontal gyrus; SMG, supramarginal gurys; SNr, substantia nigra; STG, superior temporal gyrus.

## Data availability

Raw choice and reaction time data, computational model parameter estimates, simulated data and r-maps from connectivity analysis are available via the Open Science Framework https://osf.io/fs36g/.

The following dataset was generated:

| Author(s) | Year | Dataset title | Dataset URL | Database and Identifier |
|---|---|---|---|---|
| Marcelino A, Gray O, Al-Fatly B, Gilmour W, Steele D, Kühn AA, Gilbertson T | 2022 | Explore-Exploit DBS | https://osf.io/fs36g/ | Open Science Framework, fs36g |

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
