## [Editor Report]

This paper presents valuable data from 18 patients treated with GPi DBS for dystonia using a standard RL task. Their compelling main observation is that DBS reduced the impact of value on evidence accumulation leading to more exploratory choices which was supported by fitting a dynamic decision model to the data. This work will be interesting for scientists working in fundamental and clinical neurosciences.

---

## [Decision Letter]

**Decision letter after peer review:**

Thank you for submitting your article "Pallidal neuromodulation of the explore/exploit trade-off in decision-making" for consideration by *eLife*. Your article has been reviewed by 3 peer reviewers, and the evaluation has been overseen by a Reviewing Editor and Michael Frank as the Senior Editor. The reviewers have opted to remain anonymous.

Essential revisions:

1) Generalizability:

Studying dystonia patients gives the unique opportunity to study the effects of electrical pallidal stimulation on decision-making in humans and given that dystonia primarily affects movements rather than cognition/decision-making this might also well be representative of healthy people. This (i.e. the similarity between task performance of patients and healthy people) is, however, not demonstrated in this study. In the introduction, the authors state that reward prediction error is intact in dystonic patients, but the paper that they cite for this (ref 34) is titled '… abnormal reward learning in cervical dystonia'. Furthermore, albeit clearly less pronounced than movement symptoms cognitive problems are present in dystonia patients (see Jahanshahi 2017 Movement Disorders). I would therefore recommend enrolling a healthy control group allowing to compare DBS ON and DBS OFF to healthy people.

2) Statistics:

I understand that Bayesian statistics cannot always directly be compared to non-Bayesian frequentist statistics. However, to me, the frequentist and Bayesian statistics are not consistent in this study. ANOVAs, etc are applied on subject-averages data using a p-value of 0.05 to distinguish between significant vs. non-significant results. In the Bayesian modelling analysis, the 95% HDI is computed. While this number is arbitrary (just as a p-value of 0.05) it still has a rationale to it given that in the scientific community 95% is also used for frequentist confidence intervals. Therefore, I think that 95% would be the most consistent choice here. However, none of the model parameters differ between ON vs. OFF regarding the 95% HDIs, since they overlap with 0 (see 'Contrast' in table 1). Especially the decision threshold and drift rate scaling parameter HDIs have a large overlap with 0, but they are still interpreted as significant based on the Bayes factor. The Bayes factor, however, is not used for the behavioral analyses. For example, there are no effects of DBS on decision times, but at the computational level, several parameters (which predict the decision time) are affected. I think for the sake of consistency of analyses within the paper the statistics of the Bayesian analyses should rely on the 95% HDI.

3) Connectome correlation analysis:

If I understand it correctly, the connectome analysis relates behavioral effects of stimulation to whole-brain networks rather than just local effects in the pallidum by testing whether patients who showed stronger effects of stimulation have electrodes that are closer to connections with different brain areas. In the abstract, the results of this analysis are reported as "… was predicted by the degree of functional connectivity between the stimulating electrode and prefrontal and sensorimotor cortices". In the discussion, it is stated that "…DBS-induced enhanced exploration correlated with the functional connectivity of the stimulation volume in the GPI to frontal cortical regions identified previously in functional imaging studies of explore-exploit decision making … The exploration-enhancing effects of GPI-DBS in our study were predicted by functional connectivity to brain regions whose neurons encode uncertainty [27] and predict behavioural switching[430 29, 30]". However, figure 4 essentially shows that almost the whole brain correlates with inter-individual differences in behavior reaching correlation coefficients as strong as -0.7 e.g. lower brain stem, cerebellum, and occipital cortex, none of which are mentioned in the paper. To me, it seems that there are correlations with very large and very distributed cortical areas rather than with specific areas in the prefrontal and sensorimotor cortex as stated in the paper.

Related to this point: The variable used for the connectomic correlation analysis is not the same variable that was affected by DBS in the statistical analysis. The statistical analysis found that P(explore) differed between DBS ON vs OFF irrespective of the session. Instead the "maximum within-session increase in P(Explore) DBS-ON – P(Explore ) DBS-OFF" was used.

4) In general, could you please explain this analysis in more detail? If I understand it correctly each voxel had a value for 'connectivity' to the stimulation field and a value for 'behavioral effect' and across patients, this then gave an R-map. How was figure 4 thresholded (only the maximum positive and negative Rs are given in the color bar)? Then p-values are listed. One is 0.04 and another one is 0.009. What is the difference between the two? These values seem to reflect the correlation of similarity between the individual map with the group map and the behavioral variable, but was the correlation with the behavioral variable not already used for creating the R-map? Describing the analysis in more detail might help make it more understandable to the audience not familiar with the analysis (including me).

5) It is my understanding that high exploration (e.g. P(Explore) of 0.2) should be related to poorer task performance since the optimal strategy would always use the high-value option and only switch rarely to identify the reversal(s). Why is it then that DBS can affect exploration but not the sum of rewards if the two are related? Should DBS not affect the sum of rewards if it for example was more pronounced in its effect on P(explore)?

6) Would the authors have predicted different effects for subthalamic deep brain stimulation? The DBS effects on the GPi are mainly interpreted in terms of reduced firing rate/activity. Since the STN exerts glutamatergic innervation of the GPi, should STN suppression lead to similar results? Conversely, GPe exerts GABAergic innervation of the STN. Should GPe suppression lead to the opposite behavioral effect? Were some of the electrodes localized within or close to the GPe rather than GPi and if so, did these patients show different behavioral effects?

7) Was the OFF vs ON DBS order counterbalanced? 3 patients did not complete the task OFF, and the ON dataset was not available in another patient. Did the authors check if the DBS order was relevant for the DBS effect on P(explore)?

8) The fact that a decrease in exploration behaviour isn't correlated with a modification of reward pay-off is at odds with the original theory of exploration/exploitation balance. This should at least have been discussed in order to convince the reader of the robustness of the effect observed on the P(Explore).

9) Alternative hypotheses concerning the role of the BG on the exploration/exploitation trade-off can be proposed (habits vs goal-directed behaviour, reward-driven vs automatism, etc.). They are not ruled out by the experimental results (even if we take them for granted despite (i).

10) It seems the decrease in exploration did not lead to a decrease in overall reward-but was the learning slower in the OFF condition? Figure 5 red and blue learning curves look similar, but the model fits in Figure 6 suggest a difference in positive learning rate.

If there truly is no difference in the acquisition, I'm surprised a significant reduction in exploration didn't slow learning at all either at the beginning or post-reversal… Does this imply exploration is not necessary to perform this task accurately?

11) The authors touch briefly on the differences between directed exploration (e.g. info-seeking) and random decision noise in the Discussion section; however, it might be worth mentioning earlier that current existing work on GPi (as far as I'm aware) largely links it to overall decision noise rather than information-seeking. (Whereas potentially other areas of the pallidum, e.g. GPe, have been implicated in explicit information-seeking exploration, as per White et al., 2019 Nature Comm). Clarifying this random/directed exploration difference early in the manuscript rather than later might be helpful.

12) P.3, l.108-109 ("Previously, studies of the explore-exploit dilemma have utilised bandit tasks with high pay-out volatility and/or multiple-choice options…") implies the use of the reversal learning task is novel in the context of explore-exploit. However, two-choice, simple-reversal tasks + RL models have been used before in this context: see Zhukovsky et al. 2019 Nature, Barnes et al. 2022 BP for rodent work, Oberwelland Weiss et al. 2020 FiNs for human work. None of those studies explicitly examine the role of GPi or use RLDDM, so the novelty of the present work remains, but the implication that previous studies haven't used reversal learning is perhaps misleading and might be best rephrased.

Similarly, the claim in the introduction (p2, l45) that the neural bases of the explore/exploit tradeoff remain poorly understood is perhaps outdated. The reference is from 15 years ago, and much work has been dedicated in the interim to the neural underpinnings of explore/exploit. While certainly there is not one clear, complete circuit agreed upon for calibrating the trade-off and not one single model that's more right than all others, it could be argued that the neural mechanisms behind explore-exploit are not more poorly understood than those behind most cognitive processes.

13) Would help to briefly explain and contextualize dystonia, for a broader audience who may be unfamiliar with it.

14) I am not sure why, in the connectivity analysis, DBS stimulation volume was linked to the maximum within-session ON-OFF p(Explore) difference … why separate by session in this analysis, when p(Explore) in panel A of the same figure, and in several of the other analyses + model fits, is calculated over all three sessions? Why is the within-session difference relevant here and nowhere else?

Relatedly, I'm not sure I understand what it means for the spatial similarity measure in Figure 4A to have a value less than 0… I'm not familiar with this type of analysis so I might just not be understanding it, but if the R-Map is built from an (?) average of individual patients' connectivity maps (as per l.224-226), how is it possible that 11/14 individuals have negative spatial similarity to the R-Map?

15) To my understanding, previous neural models and accounts of the GPi activity modulating exploration were within-task (e.g. in early trials all options are available for exploration, in later trials once one option emerged as best, GPi suppressed others); whereas in the present data, the manipulation happens across tasks. Thus we're seeing differences in an overall level of exploration rather than GPi-driven fluctuations in within-task behavior-which, while still in line with previous findings, warrants some additional questions. For instance:

– Does the within-task modulation of exploration differ in the OFF and ON states? Visual inspection of Figure 3D suggests no, but hard to tell visually.

– Would a finer p(Explore) window (say, per every 10 trials rather than every 40) indicate some interesting differences in the patterns of exploratory choices between the OFF and ON states, that might not be visible with averaging?

16) It might be worth clarifying explicitly in the text that the m parameter for scaling takes the place, in some sense, of the softmax function for RL in terms of calibrating explore/exploit behavior. This is not immediately obvious to readers unfamiliar with the RLDDM, but it's a relevant detail for interpreting modelling results in terms of exploration behavior.

17) In a similar vein, perhaps also state explicitly that the m parameter and the learning rate were previously found to be negatively correlated, and what that means. This will give the reader a bit of extra context to interpret and understand figure 7.

18) It might also help to explain why the learning rates can go outside of the usual [0; 1] interval (and they're on quite a larger scale as seen in the original RLDDM paper, though I suspect that's just a consequence of using a different dataset).

19) The present model can't capture potential fluctuations m across the session. (For instance, corresponding to increased exploration tendencies in the beginning, or right after the reversal.) Pedersen et al. present an alternative to the stable-scaling model, in which parameter m, rather than remain fixed, changes through the session; this does not fit as well with the data they used, but the difference in fit was small. When comparing the best RLDDM models to use, was a non-constant scaling model also tested? (The proposed power-law function might not necessarily reflect the intuitive exploration fluctuations in a reversal-learning task, which we might expect to differ from the PST in the original RLDDM paper, so the function might require some tweaks).

20) I am not certain this is feasible within the number of trials, but would fitting the RLDDM separately to each task version to pre- and post-reversal find any differences in how parameters change in the OFF and ON states?

Not sure we would expect any differences or what they would be under the present hypothesis on GPi function; however, having a finer measure for fluctuations in exploration (either through a model-free moving window of p-Explore, as suggested earlier, or through a dynamic scaling model, or by fitting pre- and post-reversal) could provide extra insight into how exploration is affected by the DBS manipulation.

*Reviewer #2 (Recommendations for the authors):*

The robustness of the effect observed on P(explore) should be discussed.

Alternative hypotheses should be discussed also.

*Reviewer #3 (Recommendations for the authors):*

Conceptual Comments

1) It seems the decrease in exploration did not lead to a decrease in overall reward-but was the learning slower in the OFF condition? Figure 5 red and blue learning curves look similar, but the model fits in Figure 6 suggest a difference in positive learning rate.

If there truly is no difference in acquisition, I'm surprised a significant reduction in exploration didn't slow learning at all either at the beginning or post-reversal… Does this imply exploration is not necessary to perform this task accurately?

2) The authors touch briefly on the differences between directed exploration (e.g. info-seeking) and random decision noise in the Discussion section; however, it might be worth mentioning earlier that current existing work on GPi (as far as I'm aware) largely links it to overall decision noise rather than information-seeking. (Whereas potentially other areas of the pallidum, e.g. GPe, have been implicated in explicit information-seeking exploration, as per White et al., 2019 Nature Comm). Clarifying this random/directed exploration difference early in the manuscript rather than later might be helpful.

3) P.3, l.108-109 ("Previously, studies of the explore-exploit dilemma have utilised bandit tasks with high pay-out volatility and/or multiple-choice options…") implies the use of the reversal learning task is novel in the context of explore-exploit. However, two-choice, simple-reversal tasks + RL models have been used before in this context: see Zhukovsky et al. 2019 Nature, Barnes et al. 2022 BP for rodent work, Oberwelland Weiss et al. 2020 FiNs for human work. None of those studies explicitly examine the role of GPi or use RLDDM, so the novelty of the present work remains, but the implication that previous studies haven't used reversal learning is perhaps misleading and might be best rephrased.

Similarly, the claim in the introduction (p2, l45) that the neural bases of the explore/exploit tradeoff remain poorly understood is perhaps outdated. The reference is from 15 years ago, and much work has been dedicated in the interim to the neural underpinnings of explore/exploit. While certainly there is not one clear, complete circuit agreed upon for calibrating the trade-off and not one single model that's more right than all others, it could be argued that the neural mechanisms behind explore-exploit are not more poorly understood than those behind most cognitive processes.

4) Would help to briefly explain and contextualize dystonia, for a broader audience who may be unfamiliar with it.

Analysis Comments

5) I am not sure why, in the connectivity analysis, DBS stimulation volume was linked to the maximum within-session ON-OFF p(Explore) difference … why separate by session in this analysis, when p(Explore) in panel A of the same figure, and in several of the other analyses + model fits, is calculated over all three sessions? Why is the within-session difference relevant here and nowhere else?

Relatedly, I'm not sure I understand what it means for the spatial similarity measure in Figure 4A to have a value less than 0… I'm not familiar with this type of analysis so I might just not be understanding it, but if the R-Map is built from an (?) average of individual patients' connectivity maps (as per l.224-226), how is it possible that 11/14 individuals have negative spatial similarity to the R-Map?

6) To my understanding, previous neural models and accounts of the GPi activity modulating exploration were within-task (e.g. in early trials all options are available for exploration, in later trials once one option emerged as best, GPi suppressed others); whereas in the present data, the manipulation happens across tasks. Thus we're seeing differences in an overall level of exploration rather than GPi-driven fluctuations in within-task behavior-which, while still in line with previous findings, warrants some additional questions. For instance:

– Does the within-task modulation of exploration differ in the OFF and ON states? Visual inspection of Figure 3D suggests no, but hard to tell visually.

– Would a finer p(Explore) window (say, per every 10 trials rather than every 40) indicate some interesting differences in the patterns of exploratory choices between the OFF and ON states, that might not be visible with averaging?

Modelling Comments

1) It might be worth clarifying explicitly in the text that the m parameter for scaling takes the place, in some sense, of the softmax function for RL in terms of calibrating explore/exploit behavior. This is not immediately obvious to readers unfamiliar with the RLDDM, but it's a relevant detail for interpreting modelling results in terms of exploration behavior.

2) In a similar vein, perhaps also state explicitly that the m parameter and the learning rate were previously found to be negatively correlated, and what that means. This will give the reader a bit of extra context to interpret and understand figure 7.

3) It might also help to explain why the learning rates can go outside of the usual [0; 1] interval (and they're on quite a larger scale as seen in the original RLDDM paper, though I suspect that's just a consequence of using a different dataset).

4) The present model can't capture potential fluctuations m across the session. (For instance, corresponding to increased exploration tendencies in the beginning, or right after the reversal.) Pedersen et al. present an alternative to the stable-scaling model, in which parameter m, rather than remain fixed, changes through the session; this does not fit as well with the data they used, but the difference in fit was small. When comparing the best RLDDM models to use, was a non-constant scaling model also tested? (The proposed power-law function might not necessarily reflect the intuitive exploration fluctuations in a reversal-learning task, which we might expect to differ from the PST in the original RLDDM paper, so the function might require some tweaks).

5) I am not certain this is feasible within the number of trials, but would fitting the RLDDM separately to each task version to pre- and post-reversal find any differences in how parameters change in the OFF and ON states?

Not sure we would expect any differences or what they would be under the present hypothesis on GPi function; however, having a finer measure for fluctuations in exploration (either through a model-free moving window of p-Explore, as suggested earlier, or through a dynamic scaling model, or by fitting pre- and post-reversal) could provide extra insight into how exploration is affected by the DBS manipulation.

---

## [Author Response]

Essential revisions:1) Generalizability:Studying dystonia patients gives the unique opportunity to study the effects of electrical pallidal stimulation on decision-making in humans and given that dystonia primarily affects movements rather than cognition/decision-making this might also well be representative of healthy people. This (i.e. the similarity between task performance of patients and healthy people) is, however, not demonstrated in this study. In the introduction, the authors state that reward prediction error is intact in dystonic patients, but the paper that they cite for this (ref 34) is titled '… abnormal reward learning in cervical dystonia'. Furthermore, albeit clearly less pronounced than movement symptoms cognitive problems are present in dystonia patients (see Jahanshahi 2017 Movement Disorders). I would therefore recommend enrolling a healthy control group allowing to compare DBS ON and DBS OFF to healthy people.

This reviewer is correct in asserting the idea that RL learning is not normal in patients with focal dystonia (e.g. Gilbertson et al., 2019[1]). In the Discussion section, “study limitations” (p24 l624-635), of the original manuscript we emphasised this as an interpretational limitation of studying a patient group with pre-existing RL learning abnormalities. Nevertheless, for comparative purposes we have included results of testing a group (n=18) of age and sex matched controls performing the first block of the reward-reversal learning task. We found P(Explore) values were higher in the HC group when compared to the dystonia patients in the DBS-OFF condition (Figure 3 —figure supplement 2A). These results replicate those of Gilbertson et al. (2019)[1] where a similar patient group (without chronic DBS implanted) used a more exploitative decision-making strategy in the same task. RLDDM fitting to the HC data confirmed lower drift diffusion scaling parameter in the HC group compared with OFF-DBS, which explained this difference in exploration (plotted in additional Figure 6 —figure supplement 2 and Table 1 – Table Supplement 2). As we found that the same effect of DBS on this model parameter explained a similar behavioural change in the patients (See point 20 and Figure 6 —figure supplement 1), we argue that the exploration enhancement by DBS is likely to reflect a normalisation of this behaviour from a pathological, disease related heightened exploitative state. This interpretational limitation has been reinforced and expanded upon in the main revised results (page 11 l 295-305) and main discussion (p24 l624-635) sections. Figure 5 has also been modified to include overlaid HC model fitting.

2) Statistics:I understand that Bayesian statistics cannot always directly be compared to non-Bayesian frequentist statistics. However, to me, the frequentist and Bayesian statistics are not consistent in this study. ANOVAs, etc are applied on subject-averages data using a p-value of 0.05 to distinguish between significant vs. non-significant results. In the Bayesian modelling analysis, the 95% HDI is computed. While this number is arbitrary (just as a p-value of 0.05) it still has a rationale to it given that in the scientific community 95% is also used for frequentist confidence intervals. Therefore, I think that 95% would be the most consistent choice here. However, none of the model parameters differ between ON vs. OFF regarding the 95% HDIs, since they overlap with 0 (see 'Contrast' in table 1). Especially the decision threshold and drift rate scaling parameter HDIs have a large overlap with 0, but they are still interpreted as significant based on the Bayes factor. The Bayes factor, however, is not used for the behavioral analyses. For example, there are no effects of DBS on decision times, but at the computational level, several parameters (which predict the decision time) are affected. I think for the sake of consistency of analyses within the paper the statistics of the Bayesian analyses should rely on the 95% HDI.

We agree with this point, thus for the sake of consistency the RLDDM parameter estimates are expressed using the 95% HDI rather than BF in the revised manuscript. Allowing for this different interpretational framework, we did not find any significant differences in the RLDDM parameters at a group level in the ON and OFF- DBS states fitted to the task as a whole. However, allowing for the recommendation to fit the RLDDM separately to pre- and post-reversal trials (response to point 20), we did find group level differences in the parameter values using the 95% HDI cut off criteria. Unsurprisingly, given both this reappraisal of the Bayesian statistics and the refitting of the model to the behavioural data, the methods, results (page 16-19, l417-465; Table 1) and Discussion sections pertaining to RLDDM have been significantly altered from the original manuscript (including an additional Figure 6 —figure supplement 1 and Table Supplement -2 pertainting to control parameter estimates ).

3) Connectome correlation analysis:If I understand it correctly, the connectome analysis relates behavioral effects of stimulation to whole-brain networks rather than just local effects in the pallidum by testing whether patients who showed stronger effects of stimulation have electrodes that are closer to connections with different brain areas. In the abstract, the results of this analysis are reported as "… was predicted by the degree of functional connectivity between the stimulating electrode and prefrontal and sensorimotor cortices". In the discussion, it is stated that "…DBS-induced enhanced exploration correlated with the functional connectivity of the stimulation volume in the GPI to frontal cortical regions identified previously in functional imaging studies of explore-exploit decision making … The exploration-enhancing effects of GPI-DBS in our study were predicted by functional connectivity to brain regions whose neurons encode uncertainty [27] and predict behavioural switching[430 29, 30]". However, figure 4 essentially shows that almost the whole brain correlates with inter-individual differences in behavior reaching correlation coefficients as strong as -0.7 e.g. lower brain stem, cerebellum, and occipital cortex, none of which are mentioned in the paper. To me, it seems that there are correlations with very large and very distributed cortical areas rather than with specific areas in the prefrontal and sensorimotor cortex as stated in the paper.Related to this point: The variable used for the connectomic correlation analysis is not the same variable that was affected by DBS in the statistical analysis. The statistical analysis found that P(explore) differed between DBS ON vs OFF irrespective of the session. Instead the "maximum within-session increase in P(Explore) DBS-ON – P(Explore ) DBS-OFF" was used.

We agree with the reviewer that the R-map method is intended to extract a whole-brain network model that is informed by a DBS-modulated output variable, in our case P(Explore) DBS-ON – P(Explore) DBS-OFF. This model is estimated using a statistical test (Pearson correlation) that capture the voxel-wise relationship between the output variable and the seed-based connectivity strength of each voxel across the participants. Voxels can therefore receive positive or negative R values indicating the strength of the aforementioned correlation. This said, positive voxels (depicted as red-yellow in the surface projection in Figure 4) are the ones to be connected to the stimulation volume in order for the latter to implicate a greater performance in P(Explore) in DBS-ON state when compared to DBS-OFF state. On the contrary, connectivity of the stimulation volume to voxels with negative R values (blue-green) would indicate worse performance in relation to P(Explore) compared between DBS states. We therefore believe that every voxel in this statistical model (map) should be taken into consideration when trying to explain covariance of stimulation associated connectivity and DBS-related changes in P(Explore) regardless of the sign of voxel-wise R. We have adapted the following paragraphs in the discussion accordingly to moderate the interpretational value of this analysis (page 23 line 600-610).

**“**Equally, this connectivity profile included regions with little to no recognised functional role in reward-reversal learning. Therefore, caution is required in interpreting the anatomical detail of this network given the limitations of using imaging data with no individualised anatomical specificity. On a more general level, this analysis provides support to the idea that discrete functions in local brain networks (in our case the GPI), are integrated into a broader scale whole-brain network. This leads to co-variation in neural activity between regions which are not conventionally attributed a functional role in a specific motor or sensory act [2] and might explain how DBS exerts influence on remote areas beyond the stimulation site [3].**”**

Additionally, we reasoned that as DBS did not increase exploration in any one specific experimental session, that it was reasonable to select the experimental session with the largest effect of DBS in any one individual. We subtracted the difference ON-OFF P(Explore) in order to reduce the between subject variation in the P(Explore) values and isolate the additional effect of DBS on enhancing exploration. This rationale has been detailed further in the Results section (page 12 l 328-345) as this approach is similar to that used previously [3, 4].

As an additional analytical check, we have also included an R-map analysis which uses the mean change in P(Explore) ON-OFF as the behavioural regressor (page 12, l 338-344, and Figure 4 —figure supplement 1). As would be expected from averaging the P(Explore) values across the whole task (rather than taking the session with the largest difference) the range of values is smaller. Crucially, the connectivity profile remained similar to the main analysis using maximal change in P(Explore) (Supplementary Figure 1). Furthermore, the relationship between the spatial similarity and the mean change in P(Explore) survived the permutation testing (R = 0.41, p(per) = 0.004).

With respect to the comment regarding “correlations with very large and very distributed cortical areas rather than with specific areas in the prefrontal and sensorimotor cortex as stated in the paper”; we agree that the interpretational value of this connectivity analysis has limited anatomical specificity. This has been acknowledge in the revised discussion (p23 l 600-610).

4) In general, could you please explain this analysis in more detail? If I understand it correctly each voxel had a value for 'connectivity' to the stimulation field and a value for 'behavioral effect' and across patients, this then gave an R-map. How was figure 4 thresholded (only the maximum positive and negative Rs are given in the color bar)? Then p-values are listed. One is 0.04 and another one is 0.009. What is the difference between the two? These values seem to reflect the correlation of similarity between the individual map with the group map and the behavioral variable, but was the correlation with the behavioral variable not already used for creating the R-map? Describing the analysis in more detail might help make it more understandable to the audience not familiar with the analysis (including me).

The reviewer has understood correctly that the connectome analysis relates stimulation-induced change in behavioral performance with functional connectivity from stimulation site to whole-brain networks and emphasizes the network effect of DBS [5]. This analysis builds on established methods previously published [4, 6, 7]. First, based on a normative connectome, functional connectivity from active contact localization to the whole brain is calculated for each patient individually (result = one functional connectivity map for each patient). Then, for each voxel in the brain, a correlation is calculated between the individual connectivity to that voxel and the behavioural parameter (result = one R value for each voxel in the brain). The R-Map depicted in figure 4B is the result of thresholding R and visualizing it into a standard brain. It represents the “optimal” functional connectivity map from the stimulation volumes to the whole brain that allows for the greatest increase in P(Explore) based on our cohort and behavioral results. The thresholds defined for the visualization of the R-Map in Figure 4B have now been added to the figure.

In the last step, we calculate the similarity from each individual connectivity map (step 1) to the R-Map (step 2), referred to as “spatial similarity”. Figure 4A shows that increased spatial similarity to the calculated R-Map is correlated with increased p(explore) and this is statistically significant (R^2^=0.28, P=0.04). Further plotting the R value distribution derived from 1000 repermuted correlations between the behavioral effect of DBS and the R-map, we show that the probability of seeing the same correlation by chance is p < 0.009 (Figure 4C). To further test the predictive value of these results, a further cross-validation analysis would be necessary. We refrained from this analysis due to the small sample size of our cohort and the strictly complementary role of this analysis. Thus, we agree with the reviewer that the statements in the introduction and abstract referring to “prediction” should be restated. Furthermore, we have rephrased the discussion of the connectivity results (see above) and have added clarity to the results and methods parts regarding the connectivity analyses (pages 11-14, l312-364).

5) It is my understanding that high exploration (e.g. P(Explore) of 0.2) should be related to poorer task performance since the optimal strategy would always use the high-value option and only switch rarely to identify the reversal(s). Why is it then that DBS can affect exploration but not the sum of rewards if the two are related? Should DBS not affect the sum of rewards if it for example was more pronounced in its effect on P(explore)?

We agree that intuitively, with greater levels of P(Explore), the number of rewards should be correspondingly lower. However, this is only absolutely true when a deterministic payout (ie 100:0%) contingency being employed or when there is a very slow learning rate. For example asymptotic performance is achieved in 100 trials in the GPI lesion study in non-human primates by Piron et al., (2016[8]). By comparison both our patients and Healthy controls achieve this in less than 10 trials. The inverse relationship between P(Explore) and performance (here we index the probability of winning a reward as a performance metric) is likely to be less robust under probabilistic reversal conditions and a high learning rate. In order to illustrate this point, and to reassure the reader of the robustness of the main results of the paper, Figure 1 demonstrates the results of plotting the relationship between performance (reward probability) and P(Explore) both from experimental (A) and simulated RLDDM data (B).

**Author response image 1. sa2fig1:** Relationship between task performance P(Reward) and probability of exploratory choices in a session. (A) Probability of making an exploratory choice in a session plotted against the probability of obtaining a rewarding outcome – P (Reward) – experimental data. (B) Simulated data. Each data point is derived from simulated data choices of the RLDDM performing the reward-reversal learning task (n=50). The dashed line represents the linear fit of this relationship between P(explore) and P(Reward). The dotted line represents a perfect negative linear relationship with slope of -1. The average experimental change in P(Explore) ON-DBS and OFF-DBS is superimposed in red and blue dotted vertical lines with the corresponding change in P(Reward) predicted from this linear fit by the horizontal equivalent line.

For the experimental data, this confirmed the expected negative relationship (rho = -0.7 p <0.001). However, fitting a linear model to this relationship confirmed that the slope was less than -1 (-0.46) meaning that for every unit increase in P(explore) the probability of winning a reward was only marginally affected. We also ran the same analysis on the 50, simulated experimental runs (Figure 1B) of the RLDDM model data choices which replicated this relationship (rho = -0.52, p<0.001). The slope of a linear model that fitted this simulated data was -0.51. Accordingly, this predicted a reduction in P(reward) from 0.79 OFF-DBS to 0.75 using the experimentally observed changes in P(Explore). From this analysis, it is possible to see that allowing for the non-deterministic reward outcomes in our task, DBS induced increase exploration by 35% leads to corresponding 6% reduction in winning a reward. This example illustrates why task performance remains relatively unaffected by the DBS despite this robustly driving greater exploratory choice through the task.

6) Would the authors have predicted different effects for subthalamic deep brain stimulation? The DBS effects on the GPi are mainly interpreted in terms of reduced firing rate/activity. Since the STN exerts glutamatergic innervation of the GPi, should STN suppression lead to similar results? Conversely, GPe exerts GABAergic innervation of the STN. Should GPe suppression lead to the opposite behavioral effect? Were some of the electrodes localized within or close to the GPe rather than GPi and if so, did these patients show different behavioral effects?

Our main rational for considering a rate coding effect on GPI firing is the established finding of GPI firing reductions during exploratory choices in primates by Sheth et al. (2011)[9]. It would seem reasonable to suggest that a similar effect may be seen with STN DBS. We think that a more straightforward prediction of this study (which gets around the difficulties of predicting exactly what the mechanism of DBS on rate coding in the STN, something which is notoriously difficult) would be that STN stimulation may interfere with the decision boundary. This has been predicted from theoretical work by Dunovan et al. (2019)[10] and experimental work with PD patients undergoing DBS by Herz and colleagues (2018)[10]. Our modelling results showing a reduction in the drift rate scaling parameter ON-DBS are experimental support for the Dunovan model of direct pathway encoding the rate of evidence accumulation.

DBS programming had been performed in all patients prior to the study and was stable at inclusion for this study. Programming had been performed in a clinical setting with the aim to achieve best possible improvement in dystonic symptoms. We have redone figure 1A to highlight the active contacts. Interestingly, active contacts localization showed a spatial distribution that agrees with the “sweet spots” for best clinical effect described in literature (ventrolateral GPi as in Schönecker et al., (2015)[11] and subpallidal white matter as in Reich et al., (2019)[12] ). Only 2 active contacts were localized in the GPe (Patient #6 and #9, see Figure 1A). Needless to say, that mere active contacts location cannot capture the extent with which GPi or GPe is being stimulated. Hence, we have added a supplementary figure (Supplementary figure 1) showing the total volume of stimulation averaged for all patients (Supplementary Figure 1A) and a heat map to visualize how many individual stimulation volumes are overlapping in each voxel of the total stimulation volume (Supplementary Figure 1B). The figure illustrates that the GPi is the focus of electrical stimulation in our cohort. We have additionally added a supplementary method section to describe how we have extracted the latter information and created the supplementary figure (page 27, l 750-770).

7) Was the OFF vs ON DBS order counterbalanced? 3 patients did not complete the task OFF, and the ON dataset was not available in another patient. Did the authors check if the DBS order was relevant for the DBS effect on P(explore)?

Yes the order was counterbalanced. We also analysed the effect of the DBS stimulation order on the P(Explore) values. We did not find any significant effect of the order of DBS stimulation on this behavioural variable. This result has been included in the revised manuscript (p10, lines 294-299).

8) The fact that a decrease in exploration behaviour isn't correlated with a modification of reward pay-off is at odds with the original theory of exploration/exploitation balance. This should at least have been discussed in order to convince the reader of the robustness of the effect observed on the P(Explore).

Please see response to point 5.

9) Alternative hypotheses concerning the role of the BG on the exploration/exploitation trade-off can be proposed (habits vs goal-directed behaviour, reward-driven vs automatism, etc.). They are not ruled out by the experimental results (even if we take them for granted despite (i).

We acknowledge that neither the task nor the model allowed us to delineate specific contributions of the BG in the explore exploit trade off within habit or goal directed learning. This could be approached with future studies with a different task and model design (for example a mixture of pavlovian and instrumental choice trails). These are important questions but were outside of the immediate scope of this experiment’s hypothesis which was to provide experimental evidence to support the longer standing theoretical proposal of the BG in explore-exploit decision-making in a more general sense.

10) It seems the decrease in exploration did not lead to a decrease in overall reward-but was the learning slower in the OFF condition? Figure 5 red and blue learning curves look similar, but the model fits in Figure 6 suggest a difference in positive learning rate.If there truly is no difference in the acquisition, I'm surprised a significant reduction in exploration didn't slow learning at all either at the beginning or post-reversal… Does this imply exploration is not necessary to perform this task accurately?

Using the HDI values rather than BF as per the original draft, we no longer found a statistically significant difference in the RLDDM learning rates in the ON- and OFF-DBS states. However, re-analysing the pre- and post- reversal trials, as suggested (see point 20) does confirm that in the OFF DBS state, lower levels of exploration are associated with lower learning rates. This is best represented in the individual parameter estimates in Figure 6 —figure supplement 1 row E for the +ve and -ve learning rate values. In the ON-DBS state the expected increase in both learning rates is notably absent when compared to the OFF-DBS condition. This once again is in the absence of any clear performance degradation; but does support the need for exploration post-reversal in the task, as the ON-DBS behaviour appears to be closer to the physiological performance of this task in the HC group. We have not formally analysed or included this in the revised manuscript, as we felt it adds another level of interpretation complexity which is outside of the scope of the original hypothesis. We would be happy to do so if it was felt that his enhanced the revised version of the paper.

11) The authors touch briefly on the differences between directed exploration (e.g. info-seeking) and random decision noise in the Discussion section; however, it might be worth mentioning earlier that current existing work on GPi (as far as I'm aware) largely links it to overall decision noise rather than information-seeking. (Whereas potentially other areas of the pallidum, e.g. GPe, have been implicated in explicit information-seeking exploration, as per White et al., 2019 Nature Comm). Clarifying this random/directed exploration difference early in the manuscript rather than later might be helpful.

We have expanded in the introduction a specific discussion (p3-4, l116-120) around the different forms of random and directed exploration and included the White et al., reference as this is relevant to the discussion around a specific role of the GPI in random exploration.

12) P.3, l.108-109 ("Previously, studies of the explore-exploit dilemma have utilised bandit tasks with high pay-out volatility and/or multiple-choice options…") implies the use of the reversal learning task is novel in the context of explore-exploit. However, two-choice, simple-reversal tasks + RL models have been used before in this context: see Zhukovsky et al. 2019 Nature, Barnes et al. 2022 BP for rodent work, Oberwelland Weiss et al. 2020 FiNs for human work. None of those studies explicitly examine the role of GPi or use RLDDM, so the novelty of the present work remains, but the implication that previous studies haven't used reversal learning is perhaps misleading and might be best rephrased.

We are grateful for highlighting this point and associated literature. We have accordingly rephrased this section of the introduction. We have removed line 108-109.

Similarly, the claim in the introduction (p2, l45) that the neural bases of the explore/exploit tradeoff remain poorly understood is perhaps outdated. The reference is from 15 years ago, and much work has been dedicated in the interim to the neural underpinnings of explore/exploit. While certainly there is not one clear, complete circuit agreed upon for calibrating the trade-off and not one single model that's more right than all others, it could be argued that the neural mechanisms behind explore-exploit are not more poorly understood than those behind most cognitive processes.

We have re-written this part of the introduction and framed the question more specifically regarding the absence of any empirical proof of a role for the BG in the explore-exploit problem in humans.

13) Would help to briefly explain and contextualize dystonia, for a broader audience who may be unfamiliar with it.

Added to introduction (p2l 94-98):

“To date, no study has directly addressed the hypothesised role of the GPI in the human approaches to resolving explore-exploit decisions during reversal-learning. Deep Brain Stimulation (DBS) of the GPI is routinely performed as a treatment for isolated generalised dystonia and also in patients with focal and segmental dystonia if botulinum toxin treatment fails [13]. Dystonia is a movement disorder characterized by sustained or intermittent muscle contractions causing abnormal postures. In focal dystonia it is restricted toone region such as the head or neck [14]”.

14) I am not sure why, in the connectivity analysis, DBS stimulation volume was linked to the maximum within-session ON-OFF p(Explore) difference … why separate by session in this analysis, when p(Explore) in panel A of the same figure, and in several of the other analyses + model fits, is calculated over all three sessions? Why is the within-session difference relevant here and nowhere else?

Please see response to point 3.

Relatedly, I'm not sure I understand what it means for the spatial similarity measure in Figure 4A to have a value less than 0… I'm not familiar with this type of analysis so I might just not be understanding it, but if the R-Map is built from an (?) average of individual patients' connectivity maps (as per l.224-226), how is it possible that 11/14 individuals have negative spatial similarity to the R-Map?

We thank the reviewer for highlighting this point. It is essential to clarify here that the extracted spatial similarity indices (y-axis of the scatter plot) are actually Fisher-z transformed R values. We first calculated R coefficient of spatial similarity using spatial Pearson correlation between R-map and each of the 14 connectivity profiles seeding from stimulation volumes. The resulting R values were then Fisher-z transformed using the built-in matlab function *atanh* which calculate the inverse hyperbolic tangent transformation of each R value. We have changed the label of y-axis in the scatter plot of Figure 4 to “Spatial Similarity *[Fz-R]*” to indicate that the R values were actually transformed to Fisher z-scores.

15) To my understanding, previous neural models and accounts of the GPi activity modulating exploration were within-task (e.g. in early trials all options are available for exploration, in later trials once one option emerged as best, GPi suppressed others); whereas in the present data, the manipulation happens across tasks. Thus we're seeing differences in an overall level of exploration rather than GPi-driven fluctuations in within-task behavior-which, while still in line with previous findings, warrants some additional questions. For instance:– Does the within-task modulation of exploration differ in the OFF and ON states? Visual inspection of Figure 3D suggests no, but hard to tell visually.– Would a finer p(Explore) window (say, per every 10 trials rather than every 40) indicate some interesting differences in the patterns of exploratory choices between the OFF and ON states, that might not be visible with averaging?

These are important points which we have tried to address with an additional and more comprehensive analysis of the temporal evolution of exploration through the task and the influence of DBS. These are included in the revised manuscript results (p10-11, l 269-294) with additional Figure 3 —figure supplement 1 (ON-DBS versus OFF-DBS) and Figure 3 —figure supplement 2 (HC versus OFF-DBS). Re-analysing the P(Explore) values ON- and OFF -DBS across twelve 10-trial bins reveals the expected increase in P(Explore) values post reversal in contingencies and an across task effect of DBS to enhance P(Explore). This analysis replicates the same effect of DBS applied to the three experimental sessions, but again we found no specific bin post reversal at which DBS clearly exerted a greater effect. We cannot discount the possibility that this reflects the small study sample size, but equally, the pan-task effect of DBS on P(Explore) may simply reflect a short task where exploration is still prominent in the pre-reversal trials as these only become overly familiar and over-learnt towards the beginning of session 2.

16) It might be worth clarifying explicitly in the text that the m parameter for scaling takes the place, in some sense, of the softmax function for RL in terms of calibrating explore/exploit behavior. This is not immediately obvious to readers unfamiliar with the RLDDM, but it's a relevant detail for interpreting modelling results in terms of exploration behavior.

We have included this clarification in the Results section regarding the *m* parameter on page 8 (l 223-225).

“It is analogous to and closely related to the β parameter (inverse temperature function) which governs the explore-exploit trade-off in a softmax choice rule.”

17) In a similar vein, perhaps also state explicitly that the m parameter and the learning rate were previously found to be negatively correlated, and what that means. This will give the reader a bit of extra context to interpret and understand figure 7.

Given the interpretational changes to the modelling result that have arisen from point [2] and point[20] we have not explicitly stated the negative relationship between learning rate and the m parameter, in part, because this is less relevant to the main modelling results (there was no group level difference in the learning rates ON-DBS). Figure 7 was originally conceived in the context of interpreting the model parameter posterior differences using BF’s. This suggested differences in three parameters of the model which figure 7 was aimed at aiding the readers interpretation of interations between these parameters on P(Explore). Given the changes to the paper from points 2 and 20, and the simpler effect of DBS on a single model parameter (drift rate scaling parameter) this figure seemed superfluous to the revised manuscript and so has been removed from this version.

18) It might also help to explain why the learning rates can go outside of the usual [0; 1] interval (and they're on quite a larger scale as seen in the original RLDDM paper, though I suspect that's just a consequence of using a different dataset).

The reason for these values not being within the usual [0,1] interval is that the learning rates are estimated in MCMC sampling process as a gaussian; we have retained this in the posterior distribution plots and tables. To transform them back into the [0,1] interval they have to be passed through an inverse logit function eg exp(x)/1+exp(x).

19) The present model can't capture potential fluctuations m across the session. (For instance, corresponding to increased exploration tendencies in the beginning, or right after the reversal.) Pedersen et al. present an alternative to the stable-scaling model, in which parameter m, rather than remain fixed, changes through the session; this does not fit as well with the data they used, but the difference in fit was small. When comparing the best RLDDM models to use, was a non-constant scaling model also tested? (The proposed power-law function might not necessarily reflect the intuitive exploration fluctuations in a reversal-learning task, which we might expect to differ from the PST in the original RLDDM paper, so the function might require some tweaks).

We are aware of the additional model variations in the original RLDDM paper by Pederson et al.[15], including the variable (rather than fixed) a and m(v scale) versions. To our knowledge, the current version of the RLDDM toolbox, which is open source, does not allow you to implement these versions (https://nbviewer.org/github/hddm-devs/hddm). However, we have re-fitted the fixed m/ v scale model to the pre- and post-reversal trials separately, as suggested in point 20. We think these revised results provide a more nuanced understanding of the mechanism underlying our DBS effect, which gets around this implementation problem.

20) I am not certain this is feasible within the number of trials, but would fitting the RLDDM separately to each task version to pre- and post-reversal find any differences in how parameters change in the OFF and ON states?Not sure we would expect any differences or what they would be under the present hypothesis on GPi function; however, having a finer measure for fluctuations in exploration (either through a model-free moving window of p-Explore, as suggested earlier, or through a dynamic scaling model, or by fitting pre- and post-reversal) could provide extra insight into how exploration is affected by the DBS manipulation.

Thank you for raising this additional analysis suggestion. We have managed to re-fit the data to the pre-and post-reversal trials (illustrated in Figure 6 —figure supplement 1 and Table 1 – Table Supplement 1). Despite the smaller number of trials the model convergedand a significant difference in the posterior distribution in m, the drift rate scaling parameter, was identified, which was reduced in the post-reversal trials in the ON-DBS condition. This was statistically significant using the 95% HDI criteria recommended in point [2]. We think this is consistent with the observed increase in P(Explore), as a reduction in, m, reduces the influence of the difference in the Q values between the choices. Allowing for this result and the different statistical interpretation of the model posterior distributions, we have had to change the discussion accordingly (p21).

References:

1. Gilbertson, T., M. Humphries, and J.D. Steele, Maladaptive striatal plasticity and abnormal reward‐learning in cervical dystonia. European Journal of Neuroscience, 2019. 50(7): p. 3191-3204.

2. Kauvar, I.V., et al., Cortical Observation by Synchronous Multifocal Optical Sampling Reveals Widespread Population Encoding of Actions. Neuron, 2020. 107(2): p. 351-367 e19.

3. Horn, A., et al., Connectivity predicts deep brain stimulation outcome in P arkinson disease. Annals of neurology, 2017. 82(1): p. 67-78.

4. de Almeida Marcelino, A.L., et al., Subthalamic neuromodulation improves short-term motor learning in Parkinson’s disease. Brain, 2019. 142(8): p. 2198-2206.

5. Horn, A., et al., Deep brain stimulation induced normalization of the human functional connectome in Parkinson’s disease. Brain, 2019. 142(10): p. 3129-3143.

6. Boes, A.D., et al., Network localization of neurological symptoms from focal brain lesions. Brain, 2015. 138(10): p. 3061-3075.

7. Corp, D.T., et al., Network localization of cervical dystonia based on causal brain lesions. Brain, 2019. 142(6): p. 1660-1674.

8. Piron, C., et al., The globus pallidus pars interna in goal‐oriented and routine behaviors: Resolving a long‐standing paradox. Movement disorders, 2016. 31(8): p. 1146-1154.

9. Sheth, S.A., et al., Basal ganglia neurons dynamically facilitate exploration during associative learning. Journal of Neuroscience, 2011. 31(13): p. 4878-4885.

10. Dunovan, K., et al., Reward-driven changes in striatal pathway competition shape evidence evaluation in decision-making. PLoS computational biology, 2019. 15(5): p. e1006998.

11. Schönecker, T., et al., Postoperative MRI localisation of electrodes and clinical efficacy of pallidal deep brain stimulation in cervical dystonia. Journal of Neurology, Neurosurgery and Psychiatry, 2015. 86(8): p. 833-839.

12. Reich, M.M., et al., Probabilistic mapping of the antidystonic effect of pallidal neurostimulation: a multicentre imaging study. Brain, 2019. 142(5): p. 1386-1398.

13. Volkmann, J., et al., Pallidal neurostimulation in patients with medication-refractory cervical dystonia: a randomised, sham-controlled trial. The Lancet Neurology, 2014. 13(9): p. 875-884.

14. Bhatia, K.P., et al., Consensus Statement on the classification of tremors. from the task force on tremor of the International Parkinson and Movement Disorder Society. Mov Disord, 2018. 33(1): p. 75-87.

15. Pedersen, M.L., M.J. Frank, and G. Biele, The drift diffusion model as the choice rule in reinforcement learning. Psychonomic bulletin and review, 2017. 24(4): p. 1234-1251.